# *Hematodinium* sp. infection does not drive collateral disease contraction in a crustacean host

Charlotte E Davies[1], Jessica E Thomas[1], Sophie H Malkin[1], Frederico M Batista[1,2], Andrew F Rowley[1], Christopher J Coates[1]*

[1]Department of Biosciences, College of Science, Swansea University, Swansea, United Kingdom; [2]Centre for Environment, Fisheries and Aquaculture Science (Cefas), Weymouth, United Kingdom

*For correspondence:
c.j.coates@swansea.ac.uk

Competing interest: The authors declare that no competing interests exist.

**Abstract** Host, pathogen, and environment are determinants of the disease triangle, the latter being a key driver of disease outcomes and persistence within a community. The dinoflagellate genus *Hematodinium* is detrimental to crustaceans globally – considered to suppress the innate defences of hosts, making them more susceptible to co-infections. Evidence supporting immune suppression is largely anecdotal and sourced from diffuse accounts of compromised decapods. We used a population of shore crabs (*Carcinus maenas*), where *Hematodinium* sp. is endemic, to determine the extent of collateral infections across two distinct environments (open-water, semi-closed dock). Using a multi-resource approach (PCR, histology, haematology, population genetics, eDNA), we identified 162 *Hematodinium*-positive crabs and size/sex-matched these to 162 *Hematodinium*-free crabs out of 1191 analysed. Crabs were interrogated for known additional disease-causing agents; haplosporidians, microsporidians, mikrocytids, *Vibrio* spp., fungi, *Sacculina*, trematodes, and haemolymph bacterial loads. We found no significant differences in occurrence, severity, or composition of collateral infections between *Hematodinium*-positive and *Hematodinium*-free crabs at either site, but crucially, we recorded site-restricted blends of pathogens. We found no gross signs of host cell immune reactivity towards *Hematodinium* in the presence or absence of other pathogens. We contend *Hematodinium* sp. is not the proximal driver of co-infections in shore crabs, which suggests an evolutionary drive towards latency in this environmentally plastic host.

## Editor's evaluation

Davies et al. present a large-scale field survey of infection status in crabs. *Hematodinium* sp., a dinoflagellate parasite that impacts crustacean fisheries worldwide. *Hematodinium* sp., previously thought to render crabs more susceptible to other infectious agents via immunosuppression, was not found to be associated with collateral infections with other disease agents. This study, instead, presents a new framework for *Hematodinium*-crab interactions; latency of the infection and absence of host immune response may drive the endemic status of *Hematodinium* sp. infections in crustaceans.

## Introduction

Host-parasite interactions are intimate and complex – the host cannot afford to overreact and risk immediate costs such as metabolic derangement (or self-reactivity) and longer-term fitness costs, yet must maintain adequate defences to fight, and recover from, parasitic insult. Likewise, parasites tend not to be hypervirulent as acute damage compromises the host and minimises reproductive and

transmission potential – so there is a broad drive towards immune-evasion for all major groups of parasites (sometimes referred to as the immune-evasion hypothesis; reviewed by *Schmid-Hempel, 2009*). Dinoflagellates of the genus *Hematodinium* include at least three parasitic species, *H. perezi*, *H. australis,* and *Hematodinium* sp., which target a myriad of crustacean hosts, as far south as Australia (*Gornik et al., 2013*) and as far north as Greenland (*Eigemann et al., 2010*). Epizootics of *Hematodinium* spp. can devastate localised communities, fishery and aquaculture industries with langoustines, shrimp, and blue crabs representing some of the >40 known susceptible marine beasties (*Albalat et al., 2016*; *Albalat et al., 2012*; *Davies et al., 2019a*; *Li et al., 2013*; *Messick and Shields, 2000*; *Rowley et al., 2015*; *Shields et al., 2003*; *Small, 2012*; *Small et al., 2012*; *Small et al., 2006*; *Smith et al., 2015*; *Stentiford et al., 2001*; *Stentiford and Shields, 2005*; *Wilhelm and Mialhe, 1996*). Signs of infection include carapace and blood (haemolymph) discolouration from the aggressive proliferation of parasite morphotypes within the liquid and solid (hepatopancreas) tissues, and lethargy caused by metabolic exhaustion (e.g., hypoproteinaemia). The advanced colonisation of the haemolymph leads to a severe decline in the number of circulating immune cells (i.e., total haemocyte counts) and regional tissue necrosis (e.g., muscle) – conditions that are likely to be fatal (*Rowley et al., 2015*). It is the conspicuous lack of host reactivity – cellular innate immunity – that is most intriguing about this host-pathogen antibiosis. Little direct evidence supports the reported view that *Hematodinium* spp. suppress the crustacean immune defences, thereby enabling the parasite to despoil its host of resources. Furthermore, direct suppression of the host's defences would leave it vulnerable to other infectious or opportunistic agents, leading to micro-scale competition with the dinoflagellate. Several studies have characterised so-called co-infections of *Hematodinium*-positive crustaceans, including bacterial septicaemia and ciliates in tanner crabs (*Chionoecetes bairdi*; *Love et al., 1993*; *Meyers et al., 1987*), and yeast-like mycosis in edible (*Cancer pagurus*; *Smith et al., 2013*) and velvet swimming crabs (*Necora puber*; *Stentiford et al., 2003*). These co-infections elicit an immune response – leading to haemocyte-directed nodulation and melanisation events (revealed by haematology and histopathology) – but during events, *Hematodinium* are not targeted. Such observations suggest an immune-evasion strategy at least at the cellular level, rather than immune suppression. Conversely, *Li et al., 2019*; *Li et al., 2015a*; *Li et al., 2015b* presented evidence for immune activation of the Japanese blue crab (*Portunus trituberculatus*) containing *H. perezi* based on measurements of immune gene levels (mRNAs), differential protein expression in the hepatopancreas, and some enzymatic activities (e.g., phenoloxidase). These data are valuable as there are few studies on the interaction between *Hematodinium* spp. and crustacean innate immunity; however, no *Hematodinium*-derived effectors were identified.

Co-infection, whereby a single individual or species is host to multiple infections (microbes or micro/macro-parasites) is commonly observed in both terrestrial and aquatic ecosystems. Characterising the drivers of these co-infections is pertinent to both the distribution of the parasite population and in commercially and ecologically important species. In open-water fisheries and aquaculture, co-infections may be more prevalent due to a large variety of environmental reservoirs and high densities, respectively. Recently, we investigated the potential role of non-commercial, common shore crabs (*Carcinus maenas*) as potential reservoirs of disease, notably *Hematodinium* spp. (*Davies et al., 2019a*), as they are co-located with many high-value shellfish, for example, edible crabs. During our initial survey, we noted the presence of several disease conditions in addition to that caused by *Hematodinium* sp., such as the parasitic barnacle *Sacculina carcini* (*Rowley et al., 2020*).

Herein, we investigated the pervasive hypothesis that *Hematodinium* spp. leaves the host more susceptible to disease, with broad implications to both parasite and host evolutionary ecology. We looked for the presence of a diverse selection of known pathogens as agents of co-infections in equal numbers of *Hematodinium*-positive shore crabs and *Hematodinium*-free controls using haematology, histology (gill, hepatopancreas), and molecular diagnostics (PCR). Haplosporidians, microsporidians, mikrocytids, *Vibrio* spp., fungal species, *S. carcini*, paramyxids, trematodes, and bacterial counts (colony-forming units [CFUs]) were studied in crabs and water across two distinct locations to account for the putative influence of environment (e.g., habitat type) on parasite presence/diversity (*Davies et al., 2020c*; *Davies et al., 2019b*). To complement the latter, we probed environmental DNA (eDNA) from the surrounding waters of infected crabs to assess the spatial and temporal ecology of target parasites.

## Results

### Are there distinct populations of *C. maenas* at the two study sites?

Overall, 1191 crabs were sampled across the year-long survey, 603 from the Dock and 588 from the Pier (*Davies et al., 2019a*). Of these crabs, 13.6% were *Hematodinium*-positive via PCR alone, with 9.3% confirmed clinical infections using molecular and tissue diagnostics. The population analysed for the present study comprised 324 crabs; 162 *Hematodinium*-positive and 162 size/sex-matched *Hematodinium*-free 'controls' as determined by haematology, hepatopancreas and gill histology, and PCR. To establish whether crabs at either site represented distinct populations, we assessed the nucleotide diversity of the mitochondrial cytochrome *c* oxidase I subunit (COI) gene from 93 crabs collected for the February 2018 screen (n = 48/Pier; n = 45/Dock) using a 588 bp fragment as recommended by *Roman, 2004*. These 93 crabs formed part of the 1191 crabs assessed for disease.

Crabs sampled from the Dock and Pier locations yielded 18 and 19 haplotypes, respectively (GenBank accession numbers MT547783-MT547812). In total, 72 COI haplotypes were identified among the 320 individual nucleotide sequences (481 bp in length) of *C. maenas* (*Figure 1*). Eight haplotypes observed in the Dock location were unique to this site (i.e., private haplotypes), and 10 private haplotypes were observed in the Pier location. Seven haplotypes were shared between the Dock and Pier locations. Globally, the most common *C. maenas* haplotype (i.e., haplotype h1 shown in yellow in *Figure 1*) was also the most common haplotype observed in the Dock (frequency of 0.33) and Pier (frequency of 0.31) locations. For all locations, haplotype diversity (Hd) ranged from 0 to 0.933 and nucleotide diversity (π) from 0 to 0.0067 (*Appendix 1—table 1*). A similar genetic diversity (i.e., number of haplotypes, Hd and π) was observed between the Dock and Pier locations (*Appendix 1—table 2*). Significant pairwise genetic differentiation (Fst estimates between 0.604 and 0.902) was observed between European off-shelf locations (located in Iceland and Faroe Islands) and all western/northern locations (*Appendix 1—table 2*). Pairwise comparison between sites within the western/northern locations revealed low Fst values, and the large majority were non-significant (p>0.05). No significant Fst value was observed between the Dock and Pier locations, indicating that the crabs from the two sites were genetically similar (*Appendix 1—table 2*).

### Are *Hematodinium* parasites infecting crabs at the two study sites genetically distinct?

No genetic differentiation (Fst = 0.004, p=0.161) was observed between the two locations. 70 ITS-1 haplotypes were identified among the 102 individual nucleotide sequences (218–229 bp in length) of *Hematodinium* sp. (*Appendix 1—table 3*). In total, 31 and 41 haplotypes were observed in the Pier and Dock locations, respectively (*Appendix 1—figure 1*). Only two haplotypes were shared between the two locations, and the most common haplotype was present at a frequency of 0.29 in the Pier and 0.23 in the Dock. A high genetic diversity was observed in both locations with a nucleotide diversity of 0.0130 and 0.0274 in the Pier and Dock, respectively.

### Does the presence of *Hematodinium* leave crabs more susceptible to collateral infections?

Across both locations, 24.7% (40/162) of *Hematodinium*-positive crabs had one or more co-infections (*Figures 2 and 3*). In terms of *Hematodinium*-free crabs, 23.5% (38/162) had one or more infection. In the Dock and Pier locations, 27.6% (24/87) and 21.3% (16/75) of *Hematodinium*-positive crabs were co-infected, with 20.7% (18/87) and 26.7% (20/75) of the *Hematodinium*-free crabs, respectively, testing positive for these notable diseases (*Figure 2*). There were no significant differences between the number of disease agents between *Hematodinium*-positive and *Hematodinium*-free crabs, regardless of location (p=0.8967 overall, p=0.3759 Dock, p=0.5667 Pier, Fisher's exact test, two-sided, *Figure 2a–c*). In the Dock location, three out of eight co-infections were observed in crabs (*Vibrio* spp., microsporidians, and *S. carcini*, *Figure 2e*, *Figure 3*); and in the Pier location, four out of eight co-infections were observed (*Vibrio* spp., *Haplosporidium* sp., trematode parasites, and fungal species, *Figure 2f*, *Figure 3*). Overall, the *Hematodinium* load quantified from the liquid tissue using haemocytometry – number of parasites per mL haemolymph – correlated positively (p<0.0001) with the severity of infection in the gill and hepatopancreas graded 0–4 from histopathology (*Supplementary file 1*).

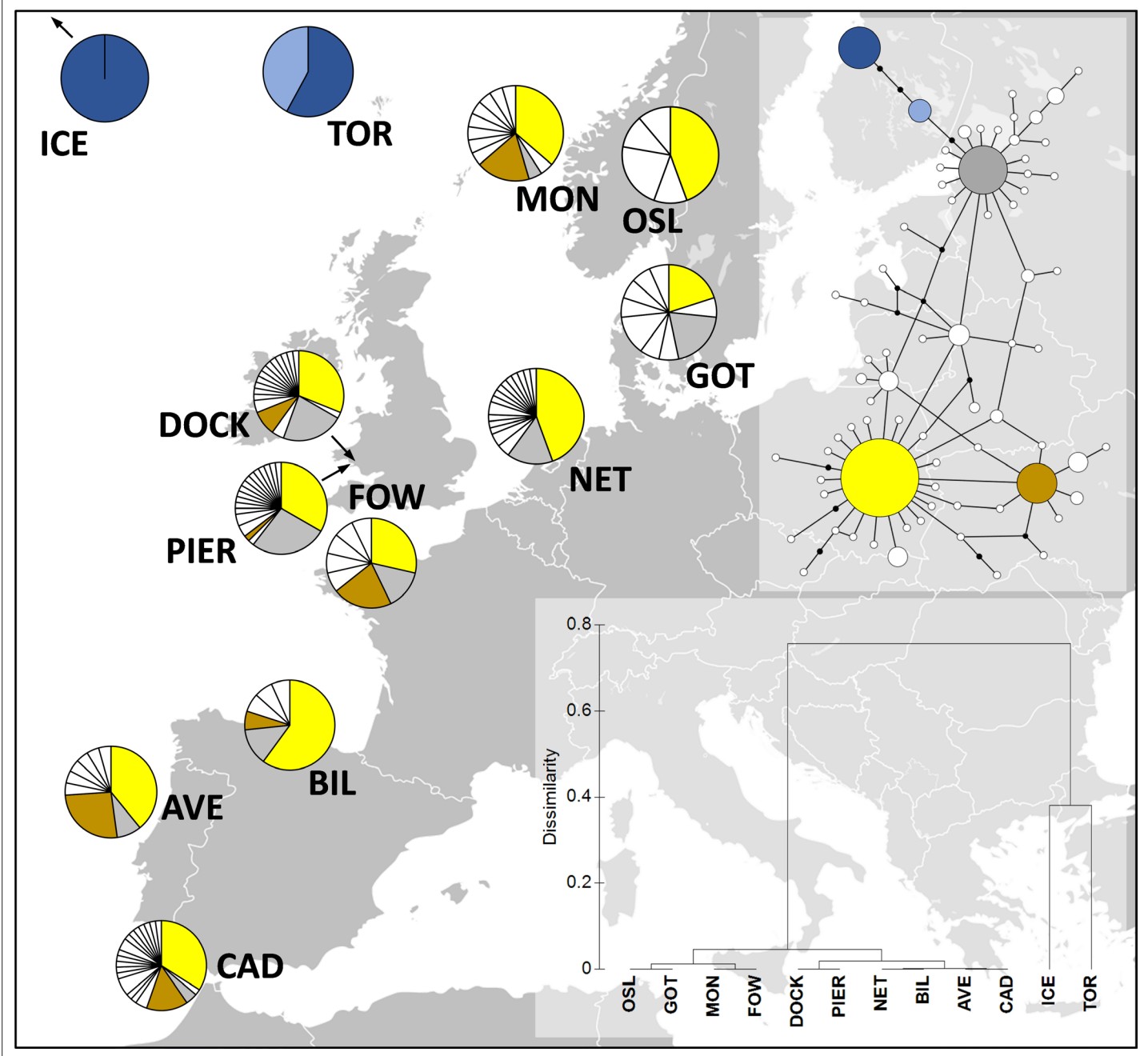

**Figure 1.** Distribution of *Carcinus maenas* haplotypes observed in the present study (Dock and Pier). At the top-right corner, a median joining haplotype network of *C. maenas* COI sequences is shown. The size of the circles of the haplotype network corresponds to haplotype frequency, and each connection represents a single-nucleotide difference. The more common haplotypes are shown in yellow (h1), brown (h6), grey (h10), dark blue (h13), and light blue (h29). The less common haplotypes are shown in white. At the bottom-right corner, a dendrogram of hierarchical clustering based on Fst values is displayed. Additional sequences were retrieved from *Darling et al., 2008*; ICE, Seltjarnarnes Iceland; TOR, Torshavn, Faroe Islands; MON, Mongstadt, Norway; OSL, Oslo, Norway; GOT, Goteborg, Sweden; NET, Den Helder, the Netherlands; FOW, Fowey, England; BIL, Bilbao, Spain; AVE, Aveiro, Portugal; CAD, Cádiz, Spain.

In terms of eDNA, we were unable to test molecularly for the presence of *Sacculina* or trematode parasites, but the remaining co-infections (six out of six) were all detected in the water (*Figure 2g*); with five out of six in the Docks (haplosporidians, microsporidians, mikrocytids, *Vibrio* spp., and fungal species; *Figure 2h*, *Figure 3*) and five out of six in the Pier; haplosporidians, paramyxids, mikrocytids, *Vibrio* spp., and fungal species (*Figure 2i*, *Figure 3*).

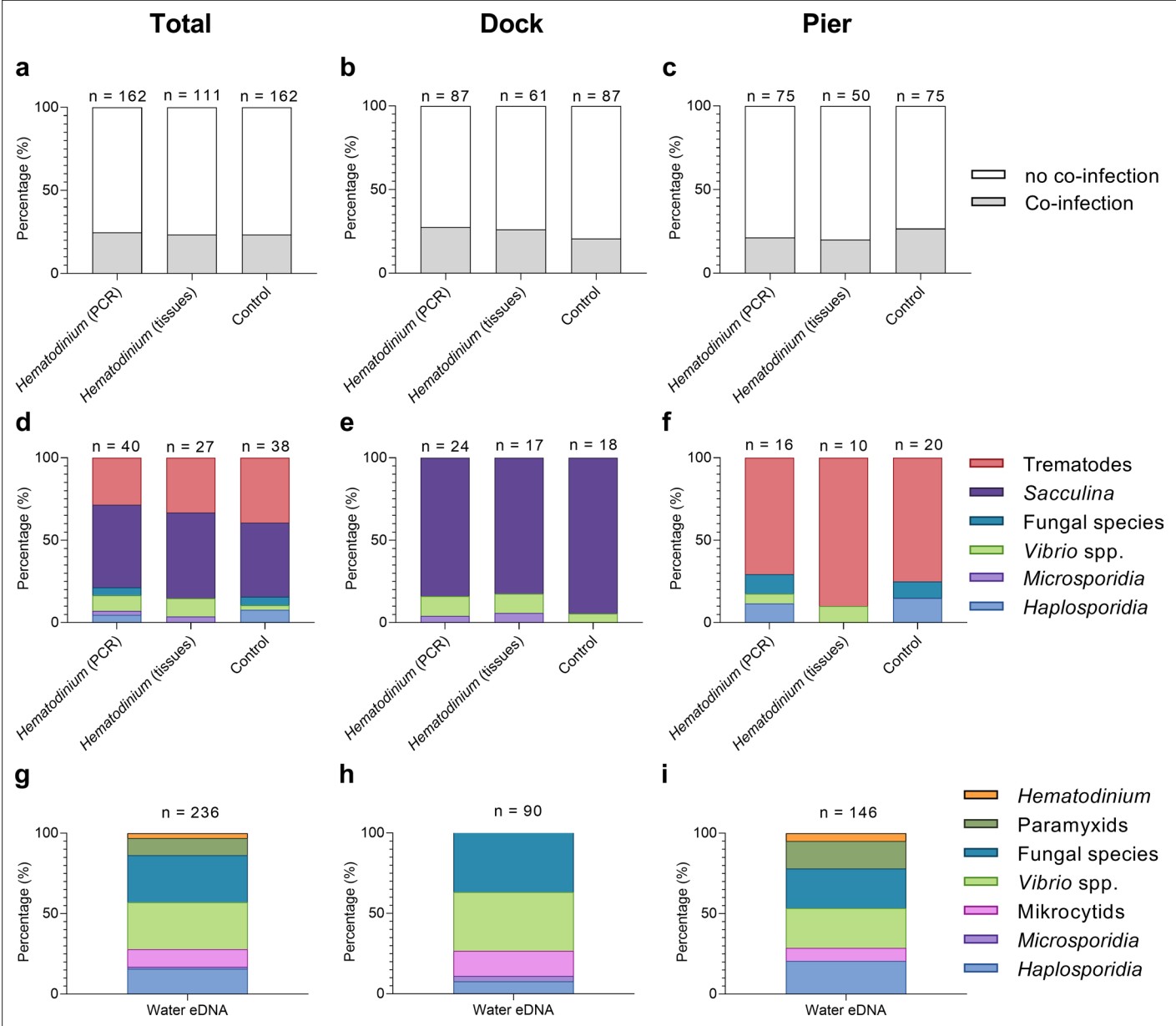

**Figure 2.** Percentage of *Hematodinium*-positive and *Hematodinium*-free ('control') crabs with and without collateral infections. Total population (**a**), Dock (**b**), and Pier (**c**) locations. Composition of co-infection(s) from those crabs which had one or more co-infections in *Hematodinium*-positive and control crabs in the total population (**d**), Dock (**e**), and Pier (**f**) locations and composition of infections, including *Hematodinium*, from seawater eDNA in total. (**g**) Dock (**h**) and Pier (**i**) locations from three filter membranes per month over 12 months. Note: trematode and *Sacculina carcini* presence were not tested for in eDNA samples but via histological examination of crab tissues only. In panels (**a–f**), crabs are represented by those testing positive for *Hematodinium* sp. via 'PCR' alone (n = 162) which includes subclinical levels of disease, and via liquid/solid 'tissue' examination (n = 111), that is, crabs showing subclinical levels of disease are excluded (middle bars).

## What factors are associated with collateral infections?

Models were run using *Hematodinium* sp. as the response variable to determine any associations between *Hematodinium* presence and co-infections (*Sacculina*, trematodes, haplosporidians, microsporidians, *Vibrio* spp., and fungal species); however, no co-infection revealed a significant relationship with *Hematodinium* sp. in the dataset overall, nor when separated by site (*Supplementary file 2*—table 1, models S4–S6; *Supplementary file 2*—table 2, models S7–S11). The number of bacterial CFUs was significantly higher in the haemolymph of *Hematodinium*-affected crabs compared

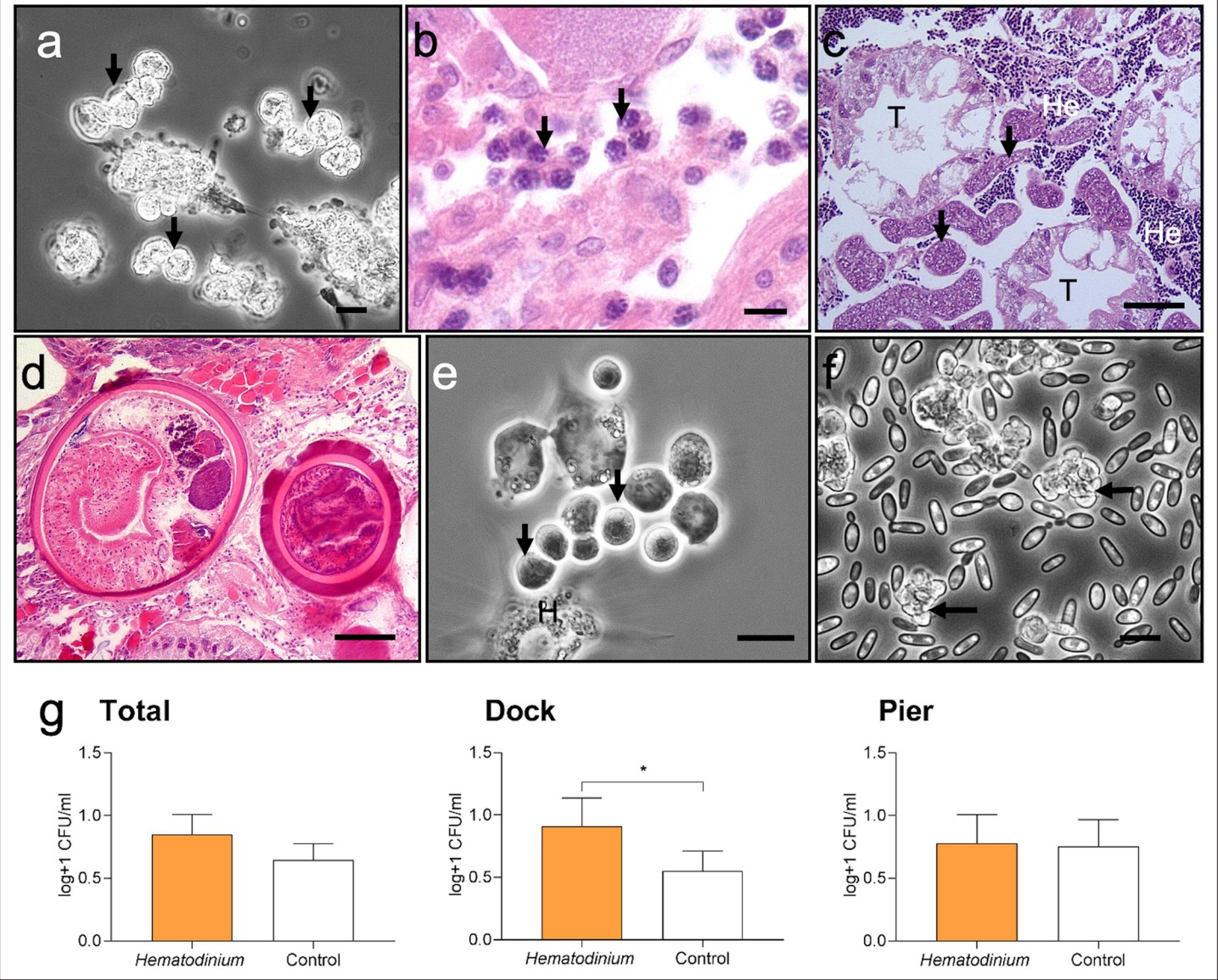

**Figure 3.** Diseases of shore crabs, *Carcinus maenas*, collected from the two reference locations. (**a, b**) Dinoflagellate parasite, *Hematodinium* (arrows), found in the haemolymph (**a**) and gonadal tissue (**b**). Scale bars = 10 µm. (**c**) Co-infected crab with the parasitic barnacle, *Sacculina carcini* (arrowheads) and *Hematodinium* (He), in the hepatopancreas. Hepatopancreatic tubules (T). Scale bar = 100 µm. (**d**) Encysted digenean trematode parasites in the hepatopancreas. Scale bar = 100 µm. (**e**) *Haplosporidium carcini* infection showing uninucleate forms (arrows) in the haemolymph. Scale bar = 10 µm. (**f**) Acute co-infection of the crab haemolymph; *Hematodinium* (arrows) and multiple yeast like fungi. Scale bar = 10 µm. (**g**) Colony-forming units (CFUs) log transformed [Y = log(y + 1)] of cultivable bacteria in haemolymph of *C. maenas* in the presence and absence of *Hematodinium* per location. Values represent mean + 95% CI, * denotes significant difference (p≤0.05).

to *Hematodinium*-free crabs, and in the Dock location only (*Figure 3g*; Mann–Whitney $U$ = 2899, p=0.0276 two-tailed).

Models were also run using the presence of one or more collateral infections as the response variable against biometric data. Model 1, the reduced model, revealed that size (carapace width [CW]) was associated with the presence of one or more co-infections (*Table 1*, model 1). Smaller crabs were significantly more likely to display co-infections compared to those that were 'disease-free' (p=0.0137, mean ± SEM: 46.26 ± 1.16 vs. 49.80 ± 0.67 mm, respectively; *Figure 4a*). *Hematodinium* presence, location, season, sex, crab colour, fouling (presence of epibionts), limb loss, and bacterial CFU number did not have a significant effect (*Figure 4e and i*; *Supplementary file 2*—table 1, model S1).

**Table 1.** Binomial logistic regression models (reduced from the full models, Supplementary file 2—table 1: model S6) testing the effects of biometric and environmental predictor variables on the overall presence of one or more co-infections.

| Model | Predictor variable | Estimate (slope) | SE | p-Value |
|---|---|---|---|---|
| **Model 1** | | | | |
| CoInfect1 ~ CW + LimbLoss | CW | –0.03368 | 0.01366 | 0.0137* |
| | Limb loss | –0.53384 | 0.33121 | 0.1070. |
| df = 320 | | | | |
| AIC: 352.13 | | | | |
| | | | | |
| **Model 2** | | | | |
| CoInfect1HEMAT ~ CW + Colour | CW | –0.03928 | 0.01905 | 0.0392* |
| + LimbLoss | Colour (orange) | –1.36238 | 0.66462 | 0.0404* |
| | Colour (yellow) | 0.39885 | 0.43242 | 0.3563 |
| df = 157 | Limb loss | –1.37492 | 0.57801 | 0.0174* |
| AIC: 168.19 | | | | |

AIC = Akaike information criterion.
SE = standard error; CW = carapace width.
Statistically significant *p≤0.05.

Model 2, the reduced model using only *Hematodinium*-positive crabs and the presence of one or more co-infections as the response variable, revealed that size (CW), crab colour, and limb loss are all associated with the presence of one or more co-infections in the crabs (*Table 1*, model 2). Smaller crabs were significantly more likely to display co-infections (p=0.0392, mean ± SEM: 46.20 ± 1.66 vs. 49.93 ± 0.96 mm, respectively; *Figure 4b*). Orange crabs were significantly less likely than green or yellow crabs to display co-infections (p=0.0404; *Figure 4f*), and those crabs which suffered the loss of one or more limbs were 2.4-fold less likely to present a co-infection than those which had not lost limbs (p=0.0174, 11.9 vs. 28.57%, respectively; *Figure 4j*). Location, season, sex, fouling (presence of epibionts), and bacterial CFU number did not have a significant effect on the presence of co-infections in *Hematodinium*-positive crabs (*Supplementary file 2*—table 1, model S2).

Using only control (*Hematodinium*-free) crabs and the presence of a 'co-infection' as the response variable produced no reduced model as the input variables, location, season, size (CW), sex, crab colour, fouling (presence of epibionts), limb loss, and CFU did not have any discernible effect (*Figure 4c, g and k*; *Supplementary file 2*—table 1, model S3).

Re-running models but restricting the crabs to those with clinical infections (n = 111) – that is, removing those that tested positive for *Hematodinium* sp. via PCR alone (n = 51) – did not yield any contradictory outcomes to those described above. Limb loss was the single variable associated significantly with co-infection detection in crabs with clinical levels of *Hematodinium* sp. (*Figure 4d, h and l*). Crucially, additional models looking at the intensity of *Hematodinium* sp. infection (no. parasites per mL haemolymph) rather than simply yes/no did not contribute significantly to the occurrence of one or more co-infection in crabs overall (p=0.335) or at either site (Pier, p=0.332, and Dock, p=0.822; *Supplementary file 2*—table 2, models S12–S14).

## Does location influence disease profiles in *C. maenas*?

In total, 80 individuals belonging to six co-infections were analysed across 78 hosts (*Source data 1*). There was no apparent significant effect of *Hematodinium* sp. presence (*F* = 0.6453, p=0.533) on co-infection number, but a significant effect of location (*F* = 94.281, p=0.001) on community structure. The non-metric multidimensional (nMDS) 2D ordination plots showed great overlap in the parasite community composition of *Hematodinium*-infected and *Hematodinium*-free crabs

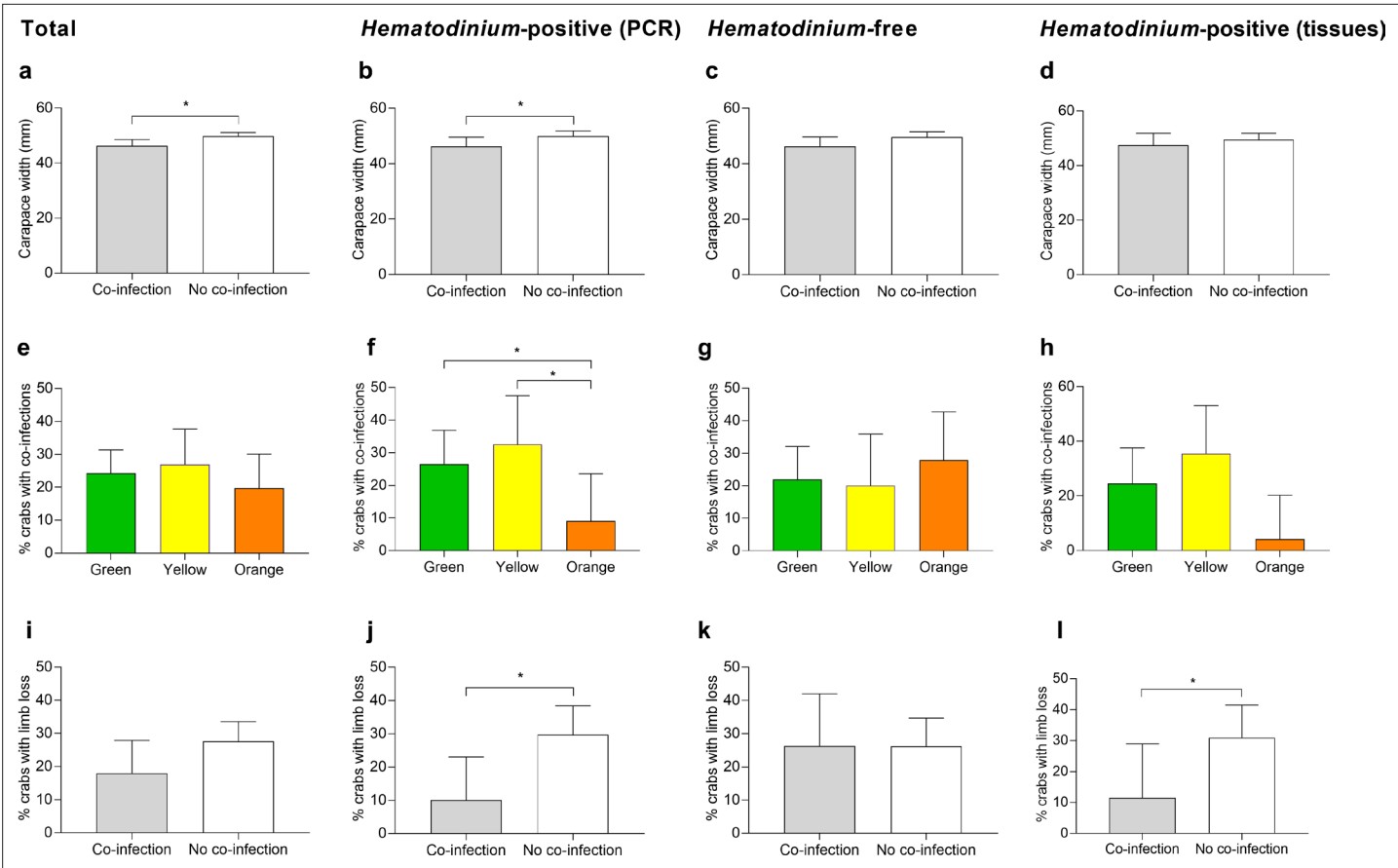

**Figure 4.** Significant factors associated with the presence of one or more co-infections. Carapace width (mm) of *C. maenas* presenting co-infections and those without in the total population (**a**), *Hematodinium*-positive by PCR (**b**), *Hematodinium*-free 'controls' (**c**), and *Hematodinium*-positive by haemolymph (tissue) inspection (**d**). Percentage of *C. maenas* presenting one or more of the co-infections according to crab colour in the total population (**e**), *Hematodinium*-positive by PCR (**f**), *Hematodinium*-free 'controls' (**g**), and *Hematodinium*-positive by tissue inspection (**h**), and percentage of *C. maenas* presenting loss of one or more limbs of the total population (**i**), *Hematodinium*-positive by PCR (**j**), *Hematodinium*-free 'controls' (**k**), and *Hematodinium*-positive by tissue inspection (**l**). Values represent mean + 95% CI, * denotes significant difference (p≤0.05).

(*Figure 5a*) but varied greatly according to Dock and Pier locations (*Figure 5b*). When crabs with subclinical levels of *Hematodinium* sp. infection were restricted from the analysis – that is, those with PCR signals alone – the overall trends were recapitulated (*Supplementary file 3*). Permutational multivariate analysis of variance (PERMANOVA) comparing the co-infection community composition between *Hematodinium* (via haemolymph quantitation)-infected and control (*Hematodinium* free) crabs yielded $R^2$ = 0.00914, therefore ~0.1% of the variation in distances is explained by the grouping of controls vs. *Hematodinium* (p=0.482; *Supplementary file 3*) in contrast to location yielding $R^2$ = 0.53818 (or nearly 54% of the variation in distances is explained by the grouping of Docks vs. Pier, p=0.001; *Supplementary file 3*). Therefore, whether or not a crab had a subclinical or progressive *Hematodinium* sp. infection did not make a difference in the composition of associated/collateral pathogens. Rather, co-infection community structure was determined by location, differing between the Dock and Pier. The differences between Dock and Pier locations were mostly driven by the presence of *Sacculina*, this being found exclusively in the Docks, as well as trematode parasites, haplosporidians, and fungal species, all of which were more abundant in the Pier location.

Temperature measurements recorded during the year-long disease survey were similar between the Pier and Dock locations (paired *t*-test, wo-sided; p=0.5329), 13.3°C ± 1.5°C and 13.6°C ± 1.6°C, respectively. Differences in salinity levels between the Pier (30.5 ± 0.9 PSU) and Dock (28.9 ± 0.6 PSU) were subtle but statistically significant (Wilcoxon, matched pairs; p=0.0425).

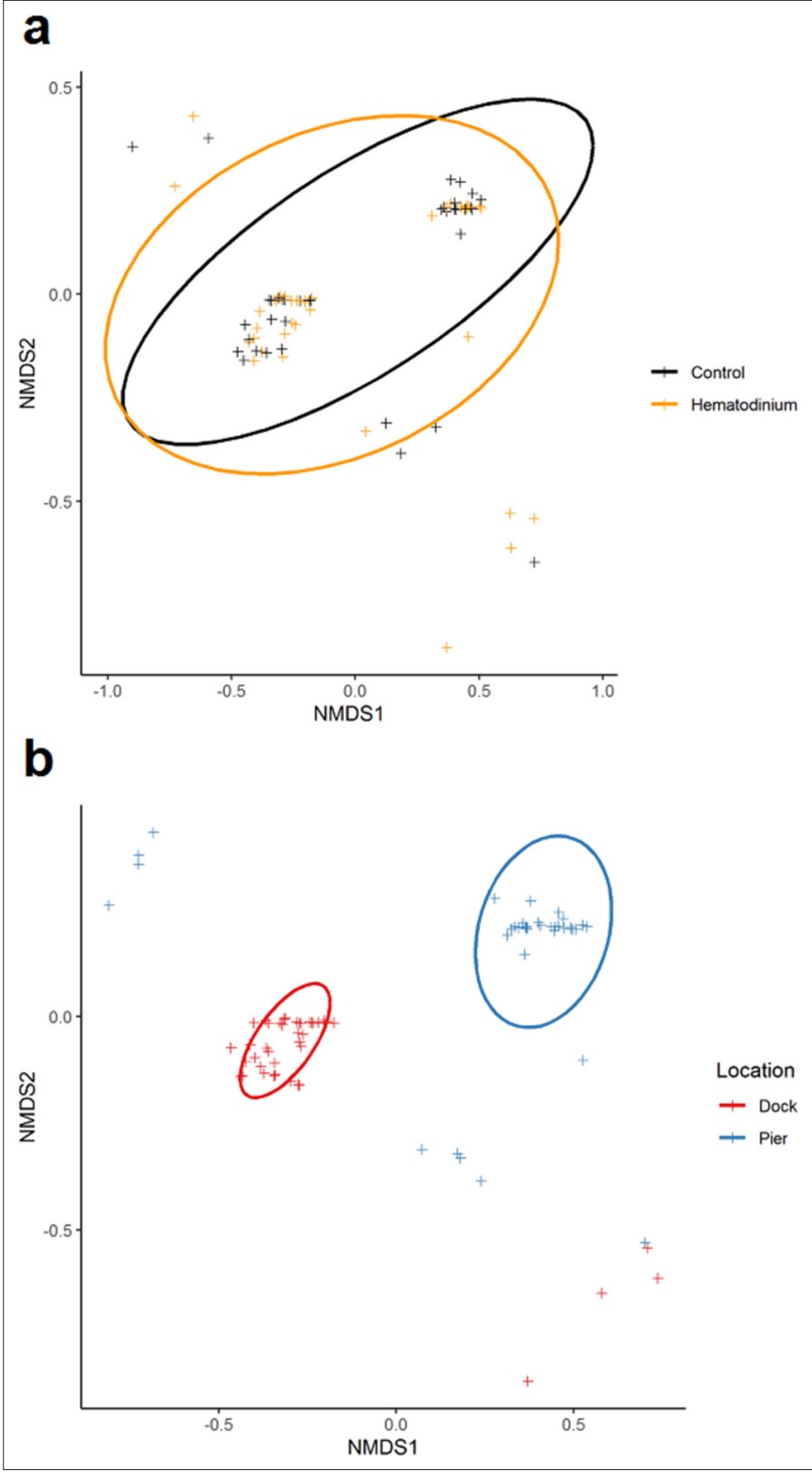

**Figure 5.** Non-metric multidimensional (nMDS) ordinations of parasite (co-infection) community structure.
(**a**) nMDS ordination co-infection/parasite (haplosporidians, microsporidians, *Vibrio* spp., fungal species,
*Sacculina carcini,* and trematodes) community structure in crabs that were *Hematodinium* sp. positive (orange)
and *Hematodinium* sp. free (black – control). (**b**) nMDS ordination co-infection/parasite (haplosporidians,

*Figure 5 continued on next page*

*Figure 5 continued*

microsporidians, *Vibrio* spp., fungal species, *S. carcini,* and trematodes) community structure in crabs from the Dock (red) and Pier (blue) locations. Analyses were done using square-root transformation of species' abundances and Bray–Curtis similarity. Each point denotes an individual crab with one or more co-infections.

### Does the crab cellular immune system respond to *Hematodinium* sp.?

We found no evidence of crab haemocyte reactivity towards *Hematodinium* sp. in the absence or presence of other disease-causing agents (n = 162; **Figure 6**) either by observing haemolymph freshly withdrawn from the haemocoel using phase contrast microscopy (**Figure 6a**) or tissue histopathology (e.g., gills and hepatopancreas). Ostensibly, crab haemocytes recognised and responded to other pathogens (**Figure 6b**) and damaged host tissues (**Figure 6c**), regardless of *Hematodinium* sp.

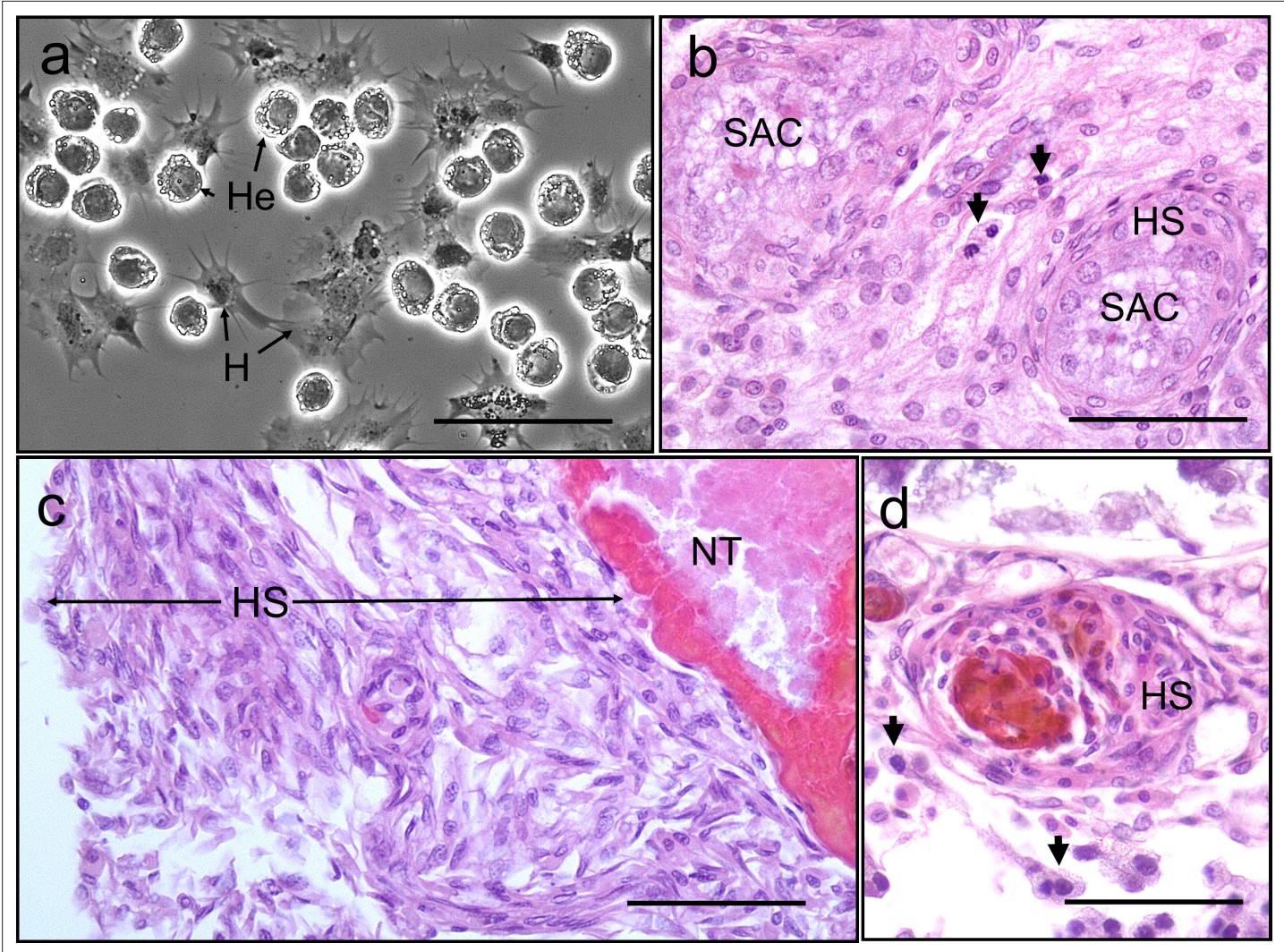

**Figure 6.** Interaction between *Hematodinium* and cellular defences of the shore crab, *Carcinus maenas*. (**a**). Phase-contrast micrograph of living cells including *Hematodinium* trophonts (He) and host's haemocytes (H). Note the lack of contact and interaction between the trophonts and these immune cells. (**b**) Host reaction in a crab with co-infection with *Hematodinium* and the rhizocephalan parasite, *Sacculina carcini*, with ensheathment by haemocytes (HS) around rootlets of *S. carcini* (SAC) in the hepatopancreas. Note that the trophonts of *Hematodinium* (unlabelled arrows) escape incorporation into the sheath around the rhizocephalan. (**c**) Infiltration and encapsulation of necrotic tissue (NT) in the hepatopancreas of a crab with a severe *Hematodinium* infection. Note that despite large numbers of this parasite in the adjacent tissues (out of shot) they do not become enmeshed within the large haemocyte sheath (HS). (**d**) Cellular response of haemocyte ensheathment (HS) around unknown debris. Note the large numbers of *Hematodinium* (unlabelled arrows) in the surrounding connective tissue but not within the haemocyte sheath. Scale bars = 50 μm.

presence. In fact, even when haemocytes infiltrated tissues in large numbers, the resident *Hematodinium* sp. were not caught up in the ensheathment process (*Figure 6d*).

Additionally, we performed in vitro phagocytosis assays using *Hematodinium* sp. isolated from a heavily infected donor crab and haemocytes from apparent disease-free crabs (n = 6). There were no gross or discernible signs of haemocyte-driven phagocytosis, degranulation, or capsule/nodule formation over the 2 hr period.

## Discussion

### *Hematodinium*-decapod antibiosis

*Hematodinium* spp. outbreaks can wreak havoc on blue crab populations in the USA (*Messick, 1994*; *Messick and Shields, 2000*; *Shields et al., 2003*), cultured decapods in China (*Huang et al., 2021*; *Li et al., 2021*; *Coates and Rowley, 2022*), and represent a persistent scourge on langoustine fisheries in Scotland (*Albalat et al., 2016*; *Albalat et al., 2012*). Although the infectivity and pathobiology of *Hematodinium* spp. in these hosts are well characterised (*Small et al., 2006*), outside of commercial settings, their role(s) as ecological regulators of crustacean populations are largely overlooked. To address this knowledge gap, we examined whether *Hematodinium* sp. infection is a determinant of co-infection health-related decline in the non-commercial shore crab *C. maenas* across two sites (semi-closed Dock vs. open-water Pier). *Davies et al., 2019a* recognised a 13.6% prevalence of *Hematodinium* sp. among these crabs using targeted PCR and reported that no significant differences in the spatial or temporal profiles of the disease between the two sites existed. Using these samples, we probed beyond *Hematodinium* sp. for the presence and diversity of known macro- and microparasites among crabs (e.g., *S. carcini* and haplosporidians, respectively) and the surrounding waters of either location using eDNA. In the host, we identified six out of the eight alternative diseases – four via both molecular and histopathology screening, and a further two via the latter technique. Data from crabs and eDNA attributed the variation in collateral infection composition to location, and not to the presence of *Hematodinium* sp. Strikingly, we found no evidence to suggest that *Hematodinium*-positive animals were more likely to harbour any one of the disease targets when compared to those diagnosed *Hematodinium*-free (*Figure 2*). This outcome is the same across, and within, both sites and is independent of the severity of *Hematodinium* sp. presence. When samples were decoupled from *Hematodinium* sp. data, site-restricted blends of parasites were obvious (*Table 2*, *Figure 2*). For example, *Hematodinium*-positive crabs from the Dock contained significantly higher levels of cultivable bacterial CFUs in the haemolymph when compared to *Hematodinium*-free animals, but this was not the case at the Pier, or when both sites were combined (*Figure 3*). The parasitic castrator, *S. carcini*, was found exclusively in the Dock site and those crabs specifically contained very high levels of CFUs, which we determined previously to be pathognomonic of *S. carcini* infection (and not *Hematodinium* sp.) in this species (*Rowley et al., 2020*).

Several articles, including the expansive review by *Stentiford and Shields, 2005*, postulate that *Hematodinium* spp. suppression of the immune response of their crustacean hosts is the most likely

**Table 2.** Detection of pathogens and parasites across sites.

| Pathogen | C. maenas | | Seawater eDNA | |
|---|---|---|---|---|
| | Dock | Pier | Dock | Pier |
| Haplosporidia | X | ✓ | ✓ | ✓ |
| Microsporidia | ✓ | X | ✓ | X |
| Mikrocytids | X | X | ✓ | ✓ |
| Paramyxids | X | X | X | ✓ |
| *Vibrio* spp. | ✓ | ✓ | ✓ | ✓ |
| Fungal species | X | ✓ | ✓ | ✓ |
| Trematode parasites | X | ✓ | NA | NA |
| *Sacculina* sp. | ✓ | X | NA | NA |

explanation for developing co-occurring secondary/opportunistic infections, including septicaemia and ciliate infections in tanner crabs (*C. bairdi*; *Love et al., 1993*; *Meyers et al., 1987*) and yeast-like infections in edible crabs (*C. pagurus*), velvet swimming crabs (*N. puber*), and shore crabs (*C. maenas*; *Smith et al., 2013*; *Stentiford et al., 2003*). Crabs are compromised to a certain extent by the presence of *Hematodinium* spp., and as the infection progresses, the host's tissues and resources are replaced with the accumulating dinoflagellate burden, yet there is no mechanistic evidence to suggest that the parasite is directly suppressing the immune system. Animals can be weakened in the absence of immunosuppression. *Li et al., 2019* describe differential protein expression in the hepatopancreas of the gazami crab *P. trituberculatus* when parasitised by *H. perezi* in laboratory settings, including the 'downregulation' of factors associated with broad-spectrum pathogen recognition. The authors contend that this is evidence of immune suppression – unfortunately, no *Hematodinium*-derived factors have been identified. Regardless of whether *Hematodinium* spp. suppress and/or evade immunity, our data reveal that the crab host is not at an increased risk of contracting another infection.

Based on these data, it is possible that *Hematodinium* sp. is an immune evader in shore crabs as its presence did not provoke cellular immunity or increase the likelihood of other diseases becoming established. *Stentiford et al., 2003* and *Stentiford and Shields, 2005* do note that there was surprisingly little evidence of host reactivity towards *Hematodinium* in crabs co-infected with yeast-like microbes. Mycosis is a rare event in the shore crabs studied here (<0.3%; *Davies et al., 2020b*). Two crabs were harbouring both yeast-like and *Hematodinium* sp. microbes, but haemocyte-derived nodulation and phagocytosis were restricted to the fungus alone.

Phagocytosis and encapsulation/nodule formation (a process where haemocytes wall off would-be colonisers) are the main cellular immune responses in invertebrates (*Ratcliffe et al., 1985*; *Coates et al., 2022*). Direct observation of live haemocytes revealed that even in cases where there are large numbers of free *Hematodinium* in circulation the haemocytes failed to recognise these as foreign. Similarly, nodules and capsules seen histologically in the tissues of *Hematodinium*-positive crabs did not contain such parasites, suggesting an active mechanism to avoid accidental incorporation into these defensive structures, perhaps molecular mimicry or concealment of surface ligands in a similar manner to entomopathogenic fungi in insect systems (reviewed by *Butt et al., 2016*). Preliminary laboratory-based studies have revealed that *C. pagurus* with modest *Hematodinium* infections cleared bacteria with similar dynamics to those free from such infections (*Smith and Rowley, 2015*), implying that alleged immune suppression by *Hematodinium* has no effect on susceptibility to unrelated infections at least in early-mid phase. Late infections by *Hematodinium* cause a marked reduction in defensive cells in circulation (*Smith and Rowley, 2015*), and these authors ascribed this to a side effect of metabolic exhaustion rather than targeted inhibition of haematopoiesis. Conversely, *Li et al., 2015a*; *Li et al., 2015b* presented evidence for parasite detection (immune recognition) and immune suppression in the *P. trituberculatus* containing *Hematodinium* sp., based on measurements of candidate immune gene expression (mRNAs) and some enzymatic activities linked to defence (e.g., phenoloxidase). Regarding these studies, it is noteworthy that the selected immune genes were not expressed/suppressed consistently across the 8-day experimental period, there was a lack of correlation between haemograms and enzymic activities (no distinction between active and total phenoloxidase activities), protein levels were not quantitated so it is unclear if increased mRNAs led to more protein, and the mode of crab inoculation itself is likely to induce at least a localised inflammatory response. Subsequently, the authors suggested that *P. trituberculatus* recognise the presence of *Hematodinium* sp. and employ oxidising/nitrosative radicals ($O_2^-$ and NO) and miRNAs to thwart the parasite (*Li et al., 2018*; *Li et al., 2016*). From our data, we cannot rule out the possibility that humoral (soluble)-mediated defences are involved in anti-*Hematodinium* immunity.

Our study showed that the presence of one or more collateral infections overall (regardless of *Hematodinium* sp. presence or not) was characterised by the size of the animal. We also determined CW (size), colouration, and limb loss to be significant predictor variables for detecting one or more co-infections in *Hematodinium*-positive crabs (*Table 1*, *Figure 4*), but not in *Hematodinium*-free crabs. The fact that autotomised crabs (i.e., those with missing limbs) are less likely to develop co-infections is intriguing – suggesting the damage/trauma may temporarily 'prime' the immune system. There is some evidence that autotomy and ablation initiate stress responses in another crab, *Eriocheir sinensis*, linked to differential haemocyte counts (*Yang et al., 2018*). Size alone – smaller

CW – was the common predictor variable for co-infection occurrence among all crabs screened. This is to be expected as juvenile crustaceans are known to be at higher risk of contracting disease when compared to their older counterparts (*Ashby and Bruns, 2018*). Larger crustaceans have longer moult increments (therefore moult less often) than smaller, younger individuals (*Castro and Angell, 2000*), giving more time for co-infections to manifest. Indeed, it is postulated that *Hematodinium* zoospores use the soft cuticle found in newly moulted crabs as a portal of entry to the haemocoel. Injury or breaching of the cuticle can act as a portal of entry for microparasites (*Davies et al., 2015*; *Coates et al., 2022*).

Co-infection incidence or composition did not follow established seasonal patterns associated with *Hematodinium* dynamics in this host (*Supplementary file 2*—table 1) – high severity and low prevalence in the winter, followed by low severity and high prevalence in the spring (*Davies et al., 2019a*). No temporal patterns of collateral infection cases were found in either *Hematodinium*-positive or *Hematodinium*-free animals. In addition, we found no evidence of genetic differentiation between crabs – and their resident *Hematodinium* ecotypes – sampled from the Dock and Pier, with both locations exhibiting similar genetic heterogeneity (*Figure 1*, *Appendix 1—Tables 1–3*, *Appendix 1— figure 1*). The lack of substantive genetic diversity of host and *Hematodinium* between the sites is unlikely to account for the different disease profiles recorded, as such; the evidence implies that disease contraction in shore crabs depends on their environment.

## Environment-driven contraction of disease

Environmental conditions of the sites tend to differ – most likely due to the hydrology of a semi-enclosed Dock vs. an open-water Pier. As part of our sampling regime, we recorded temperature and salinity, while temperature did not differ significantly between each site over the 12-month screen, salinity did. Changes in such environmental factors, as well as pH, nitrogenous wastes, and xenobiotics, can influence both host and pathogen, and the incidence of disease (reviewed by *Coates and Söderhäll, 2021*). Nonetheless, both experimental sites have similar incidence of *Hematodinium* sp. infections as well as comparable temporal dynamics of the parasite life cycle (*Davies et al., 2019a*). The life history of this parasite involves direct transmission of disease resulting from moribund crabs releasing motile zoospores into the water column to infect other susceptible crustaceans (*Stentiford and Shields, 2005*), and there is no known non-crustacean reservoir of this disease. The Mumbles Pier location supported a higher diversity of disease in terms of eDNA as well as within the crabs themselves, notably, two new species of Haplosporidia (*Davies et al., 2020a*). Studies of historical data in Swansea Bay, where the Mumbles Pier location is based, reveal persistence of benthic fauna associations as a heavily modified waterbody bearing the 'historical scars' of nearby heavy industry and limited sewage treatment (*Callaway, 2016*). Despite this, the area surrounding Mumbles Pier showed a significantly higher species richness in benthic fauna than other locations across the Bay (*Callaway, 2016*). There are few studies on species or biodiversity in the Prince of Wales Dock, but anecdotal observations suggest a sludge-like benthic environment with a large community of *C. maenas* and mussels, *Mytilus edulis*, compared with the much more diverse Pier (*Callaway, 2016*; *Powell-Jennings and Callaway, 2018*).

The profile of other microbes/parasites differed between the two sites – notably with *S. carcini* in the Dock and trematode infestations at the Pier. A possible explanation of the differences in these diseases between the open-water site (Pier at Mumbles Head) and semi-enclosed dock site relates to the presence of reservoirs and/or alternate hosts of disease as well as physical properties. For example, the unidentified digenean trematode parasites seen in the hepatopancreas of crabs take the form of encysted metacercarial stages (*Figure 3d*). Trematodes have multi-host life cycles, and predation of infected crabs by sea birds results in this definitive host becoming infected, subsequently releasing infective stages in their faeces that infect various littorinid molluscs as the first intermediate host (*Blakeslee et al., 2015*). Presumably, the putative absence of grazing littorinids in the semi-enclosed Dock breaks the infection cycle despite the presence of both shore crabs and sea birds in this site. The limited water flow in the Dock site probably favours the transmission of *S. carcini* (water may stagnate and permit parasites to accumulate), whereas the tidal flow at Mumbles reduces the chance of infectious stages contacting uninfected crabs.

## Concluding remarks

Species of the parasitic dinoflagellate genus *Hematodinium* represent a substantive, yet often over-looked, threat to populations of commercially important crustaceans globally. Our work described herein takes a major step forward in our understanding of crab-*Hematodinium* antibiosis.

Pre-existing *Hematodinium* sp. infection is not a determinant for collateral disease contraction in shore crabs. No significant differences were detected with respect to 'co-infection' levels between *Hematodinium*-positive vs. *Hematodinium*-free crabs overall, or at either geographically close site. Clear site-specific blends of parasites were found in the hosts, regardless of *Hematodinium* presence/absence, and in the surrounding waters. Herein, binomial logistic regression models revealed CW (small), and not season or sex, as a significant predictor variable of co-infections overall. This contrasts with our previous study (*Davies et al., 2019a*) in which we determined seasonality and sex, but not size, as the overall key predictor variables of *Hematodinium* sp. in crabs. If *Hematodinium* was a determinant for co-infections, we should see a seasonal pattern, or sex bias, but we do not. Crucially, the severity of *Hematodinium* sp. colonisation of tissues – either solid or liquid – is not attributed to collateral infections either. Therefore, we contest that co-infection occurrence is decoupled from *Hematodinium* sp., with location being the main determinant (habitat or surrounding diversity; *Davies et al., 2019b*, *Davies et al., 2020c*).

We did not observe immune cell reactivity in vitro or in vivo – phagocytosis, encapsulation, or melanisation – towards *Hematodinium* sp. in crabs in the absence and presence of other disease-causing agents or damaged tissue. If *Hematodinium* sp. was supressing the immune system of crabs, we would expect to see more alternative opportunistic infections (we do not) and/or a reduced capacity of the host to react to other agents (we do not). Haemocyte-driven responses remain intact in early infections, but never target *Hematodinium*. We consider *Hematodinium* sp. to be a candidate immune evader of shore crab cellular defences. This does not rule out a potential immune-suppression mechanism at the molecular level.

# Materials and methods

## Sample collection

The study took place off the South Wales coast, UK, at two distinct locations described in *Davies et al., 2019a*. The first location, a semi-closed Prince of Wales Dock, Swansea, and the second, intertidal Mumbles Pier (referred to forthwith as Dock and Pier). For 12 months from November 2017 to October 2018, the shore crab population (n = 1191) and seawater for environmental DNA analysis were surveyed at both locations. On each sampling day, water temperature and salinity were recorded using a YSI 650 MDS multi-parameter display system. A minimum of 48 crabs per site per month were surveyed to achieve >80% desired statistical power ($\alpha$ = 0.05, two-sided test). Laboratory regime, water filtration (2 L water [three technical replicates = 6 L] per location per month using an initial 200 µm nylon mesh followed by 0.45 µm PVDF [Durapore] membrane), histopathology, and DNA extraction/quantification followed the procedures of *Davies et al., 2019a*. Environmental (water) samples were collected alongside crabs at the Dock location and via boat for the Pier. In addition, the present study included quantification of bacterial CFUs, which were determined by spreading 200 µL 1:1 haemolymph:sterile 3% NaCl solution (w/v) onto tryptone soya agar (TSA) plates supplemented with 2% NaCl (two technical replicates were performed per biological replicate). Plates were incubated at 25°C for 48 hr and CFUs counted. The bacterial load of the haemolymph is expressed as CFUs/mL.

## PCR-based approaches and sequencing conditions

All PCR reactions were carried out in 25 µL total reaction volumes using 2X Master Mix (New England Biolabs), oligonucleotide primers synthesised by Eurofins (Ebersberg, Germany), 1 µL DNA (ca. 50–200 ng for haemolymph and 3–80 ng for water eDNA), and performed on a Bio-Rad T100 PCR thermal cycler. Products derived from PCR were visualised on a 2% agarose/TBE gel with Green-Safe premium nucleic acid stain (NZYTech, Portugal). For primary diagnostics, general *Hematodinium* primers targeting a highly variable 18S rRNA gene region (Hemat-F-1487 and Hemat-R- 1654; *Supplementary file 2*—table 1) were used to verify the presence of any *Hematodinium* (see *Davies et al., 2019a* for full details). Proceeding this, a control group of equal number, size/sex/location-matched

*Hematodinium*-free crabs (n = 162) were chosen, and both groups were subjected to a series of targeted PCRs to determine the presence of haplosporidians, microsporidians, mikrocytids, paramyxids, *Vibrio* spp., and fungal species (*Appendix 2—table 1*). Positive amplicons were purified using HT ExoSAP-IT Fast High-Throughput PCR Product Cleanup (Thermo Fisher Scientific, UK) following the manufacturer's instructions, quantified using the Qubit dsDNA High Sensitivity Kit and Fluorometer (Invitrogen, USA), and sequenced using Sanger's method by Eurofins.

All sequences have been deposited in the GenBank database under the accession numbers MN846355–MN846359 (from *C. maenas* haemolymph DNA) and MT334463–MT334513 (seawater eDNA) for haplosporidians (*Davies et al., 2020a*); MN985606–MN985608 (seawater eDNA) and MN985609 (*C. maenas* haemolymph DNA) for microsporidians, MN985610–MN985644 (seawater eDNA) for paramyxids, MT000071–MT000098 (seawater eDNA) for mikrocytids and MT000100–MT000103 (*C. maenas* haemolymph DNA) and MT000104–MT000107 (seawater eDNA) for fungal species. Sequences for *Vibrio* spp. (<150 bp in length) were deposited in the NCBI short read archive (SRA) under accession numbers SAMN14133753 to SAMN14133757 (*C. maenas* haemolymph DNA) and SAMN14133758- SAMN14133765 (seawater eDNA; see *Supplementary file 4* for complete information).

## Population (genetic) analyses

The COI gene was amplified from crab DNA from crabs gathered in February 2018 across both survey sites (n = 100 in total, i.e., n = 50 per site) using oligonucleotides from *Roman, 2004* (*Appendix 2—table 1*). PCR reactions were carried out as described above (25 µL total volume, Q5 hot start high fidelity 2X master mix [New England Biolabs], oligonucleotide primers synthesised by Eurofins, 1 µL DNA [ca. 50–200 ng], and visualised on a 2% agarose/TBE gel). Amplicons were purified as above and sequenced using Sanger's method by Source BioScience (Nottingham, UK). Chromatograms of the nucleotide sequences were analysed using BioEdit version 7.0.9.0 (*Hall, 1999*). Sequences were aligned and trimmed using BioEdit, resulting in 93 COI sequences (n = 48 for the Pier location; n = 45 for the Dock location) with 481 bp (yielding 37 haplotypes; GenBank accession numbers MT547783-MT547812). Additionally, we included in our analyses 227 COI sequences of *C. maenas* from *Darling et al., 2008* (GenBank accession numbers FJ159008, FJ159010, FJ159012-13, FJ159015-18, FJ159020-21, FJ159023, FJ159025-36, FJ159039-44, FJ159047-52, FJ159057, FJ159059-64, FJ159069-80, FJ159084, and FJ159085) across 10 locations (see Appendix 1 for details). ARLEQUIN (version 3.11) was used to calculate the number of haplotypes, haplotype diversity, and nucleotide diversity (*Excoffier et al., 2006*; *Excoffier and Lischer, 2010*). Pairwise genetic differentiation (Fst) values using 10,000 permutations were calculated among the 12 locations using ARLEQUIN. A median joining network (*Bandelt et al., 1999*) using the 293 *C. maenas* COI nucleotide sequences was constructed using PopART version 1.7 (*Leigh and Bryant, 2015*). To visualise the genetic similarities between locations, a hierarchical clustering analysis based on Fst values with 500 random starts was performed using PRIMER v6 (*Clarke and Gorley, 2006*).

*Hematodinium* sp. nucleotide sequences (partial coverage of the ITS1 region) from infected crabs (n = 162; GenBank MN057783–MN057918) (*Davies et al., 2019a*) representing both the Pier (open) and Dock (semi-closed) locations were reassessed for genetic diversity. Sequences were inspected manually, trimmed, aligned, and those with undetermined (ambiguous) nucleotides were removed using BioEdit. In total, 102 *Hematodinium* sp. nucleotide sequences between 218 and 229 bp in length (n = 49 for the Pier location; n = 53 for the Dock location) were analysed using ARLEQUIN (taking into account insertions/deletions) as described above for *C. maenas* COI sequences.

## Tissue histology and microscopy

Haemolymph preparations from all 324 crabs were assessed for the presence of parasites and pathogens, and putative (host) cellular responses (e.g., phagocytosis). To accomplish this, haemolymph was rapidly withdrawn from the haemocoel of each crab using a 22-gauge hypodermic needle inserted into a walking leg, placed on glass slides, and examined using phase-contrast microscopy. If the sample appeared to contain parasites, a further 25 µL of fresh haemolymph was fixed 1:1 with 5% formaldehyde (v/v) in 3% NaCl (w/v) and placed on an improved Neubauer haemocytometer where *Hematodinium* morphotypes were quantified (i.e., number of parasites per mL).

In July 2021, we collected an additional 58 crabs from the Dock area, extracted haemolymph, and screened for *Hematodinium* sp. as described above. For the parasitised crabs (n = 9), we maintained the haemolymph ex vivo for up to 75 min to inspect haemocyte behaviour microscopically. In a further experiment, we isolated haemocytes from six healthy (*Hematodinium*-free) crabs (n = 3 males, n = 3 females). Haemolymph was removed and diluted 1:1 into anticoagulant (3% NaCl, 100 mM dextrose, 47 mM citric acid, 10 mM EDTA, pH 4.6) adapted from *Söderhäll and Smith, 1983*, centrifuged at 500 × *g* at 4°C for 5 min, resuspended in phagocytosis assay buffer (3% NaCl, 20 mM HEPES, pH 7.5, 10 mM CaCl$_2$, 10 mM MgCl$_2$, 5 mM KCl$_2$, and 10 mM NaHCO$_3$; *Coates et al., 2012*), before being seeded into individual wells of a sterile culture plate containing *Hematodinium* isolated from an infected donor crab (following *Smith and Rowley, 2015*). Assays were run for 2 hr at room temperature and terminated by the addition of 2.5% formaldehyde (v/v).

Tissue histology was used as the secondary tool after PCR, to screen all 324 crabs to estimate the severity of, and potential immune responses to, *Hematodinium* sp. or any collateral infection (e.g., phagocytosis, melanisation, haemocyte aggregation). Histology took place according to the methods described in *Davies et al., 2019a*. Briefly, gills and hepatopancreas/gonad were excised and fixed in Davidson's seawater fixative for 24 hr prior to their storage in 70% ethanol. Samples were processed using a Shandon automated tissue processor (Thermo Fisher Scientific, Altrincham, UK) prior to wax embedding. Blocks were cut at 5–7 µm thickness using an RM2245 microtome (Leica, Wetzlar, Germany), and sections were mounted onto glass slides using albumin-glycerol. All slides were stained with Cole's haematoxylin and eosin prior to inspection using an Olympus BX41 microscope. The gills and hepatopancreas/gonads of *Hematodinium*-positive crabs were graded (0–4) according to severity of infection following the criteria established by *Smith et al., 2015*. Briefly, a score of 0 indicates no visible infection (i.e., subclinical) despite being PCR positive, whereas a score of 4 indicates tissue replete with parasites (and few, if any, haemocytes visible).

## Statistical analyses

Binomial logistic regression models with Logit link functions (following Bernoulli distributions) were used ('MASS' package) to determine whether specific predictor variables had a significant effect on the probability of finding crabs testing positive for *Hematodinium* presence in the crab populations sampled. Models were run using *Hematodinium*-positive crabs determined via PCR alone (n = 162), and *Hematodinium*-positive crabs quantified by haemolymph screening via phase-contrast microscopy, gill/hepatopancreas via histology, and PCR (n = 111, i.e., visibly clinically infected crabs). The information theoretic approach was used for model selection and assessment of performance (*Richards, 2005*). Initial models are herein referred to as the full models. Once selected, each non-significant predictor variable from the full models was sequentially removed using the drop1 function to produce final models with increased predictive power, herein referred to as the reduced models. The drop1 function compares the initial full model with the same model minus the least significant predictor variable. If the reduced model is significantly different from the initial full model (in the case of binomial response variables, a chi-squared test is used to compare the residual sum of squares of both models), then the removed predictor variable is kept out of the new, reduced model (*Table 1*). This process continues hierarchically until a final reduced model is produced (*Zuur et al., 2009*). Full models included the input variables: *Hematodinium* (presence of parasite, 0 or 1 – based on PCR and microscopic diagnostics), location (Dock or Pier), season (winter [Dec '17, Jan '18, Feb '18], spring [Mar '18, Apr '18, May '18], summer [Jun '18, Jul '18, Aug '18], autumn [Sept '18, Oct '18, Nov '17]), CW (continuous number), sex (male or female), carapace colour (green, yellow, or orange), fouling (presence of epibionts, 0 or 1), and limb loss (0 or 1; for all full models, see *Supplementary file 2*—table 1). Additionally, linear regression was applied to the quantitative *Hematodinium* data to determine whether there was a relationship between the number of parasites visible in the liquid (haemolymph) tissue and the solid (gill, hepatopancreas) tissues of affected crabs (n = 108).

To explore the effects of location (Pier or Dock) and *Hematodinium* (0 or 1 – based on PCR and microscopic diagnostics) on the co-infection assemblage structure (based on abundances of individual infections), multivariate analysis of community composition was used. First, those crabs which suffered from one or more co-infections were subsampled (n = 78) and an unconstrained PERMANOVA was run using the 'adonis' analysis ('vegan' package). This analysis was based on Bray–Curtis dissimilarities and 999 permutations. PERMANOVA is non-parametric, based on

dissimilarities, and uses permutation to compute an F-statistic (pseudo-F). nMDS scaling using the Bray–Curtis measure on a square-root transformation of the abundance data was also used to visualise differences in community composition between groups. This transformation retains the quantitative information while down-weighing the importance of the highly abundant infections (*Clarke, 1993*).

Bacterial CFU numbers and haemocyte counts were log transformed [$Y = \log(y + 1)$], and following testing for normality, a Mann–Whitney test (unpaired) was performed to compare ranks between *Hematodinium*-positive vs. *Hematodinium*-free crabs, and infections within *Hematodinium*-positive/free crabs. All logistic models and composition analysis were run in RStudio version 1.2.1335 (2009–2019 RStudio, Inc) using R version 3.6.1. All other statistics (tests of normality, transformations, and *t*-tests or non-parametric equivalent) as well as graphics were produced using GraphPad Prism v8 for Windows (GraphPad Software, La Jolla, CA).

## Acknowledgements

This study was part-funded by the European Regional Development fund through the Ireland Wales Cooperation Programme, BLUEFISH, awarded to CJC and AFR. AFR was also part-funded by the BBSRC/NERC ARCH UK Aquaculture Initiative (BB/P017215/1), and start-up funds from Swansea University assigned to CJC were used to supplement this study. The authors thank Mr. Peter Crocombe, Ms. Charlotte Bryan, Ms. Jenna Haslam, and Ms. Emma Quinn, and boat skippers, Mr. Keith Naylor, Mr. Max Robinson, and Mr. Barry Thomas, for assistance in the laboratory and field, respectively.

## Additional information

### Funding

| Funder | Grant reference number | Author |
| --- | --- | --- |
| Interreg | BlueFish | Charlotte E Davies<br>Jessica E Thomas<br>Sophie H Malkin<br>Frederico M Batista<br>Andrew F Rowley<br>Christopher J Coates |
| Biotechnology and Biological Sciences Research Council | BB/P017215/1 | Andrew F Rowley<br>Christopher J Coates |
| Swansea University | | Christopher J Coates |

The funders had no role in study design, data collection and interpretation, or the decision to submit the work for publication.

### Author contributions

Charlotte E Davies, Data curation, Formal analysis, Investigation, Methodology, Software, Validation, Visualization, Writing – original draft, Writing – review and editing, Co-corresponding author: c.e. davies@swansea.ac.uk; Jessica E Thomas, Investigation, Methodology; Sophie H Malkin, Data curation, Investigation, Methodology, Project administration; Frederico M Batista, Data curation, Formal analysis, Investigation, Methodology, Software, Visualization; Andrew F Rowley, Data curation, Formal analysis, Funding acquisition, Investigation, Methodology, Project administration, Resources, Software, Supervision, Visualization, Writing – original draft, Writing – review and editing; Christopher J Coates, Conceptualization, Data curation, Formal analysis, Funding acquisition, Investigation, Principal/lead investigator., Methodology, Project administration, Resources, Software, Supervision, Validation, Visualization, Writing – original draft, Writing – review and editing

### Author ORCIDs

Charlotte E Davies  http://orcid.org/0000-0002-5853-1934
Christopher J Coates  http://orcid.org/0000-0002-4471-4369

## Ethics

Although work on non-cephalopod invertebrates was not regulated by the Home Office or UK legislation at the time of this study, screening C. maenas for pathogens and parasites was approved by the College of Science (Swansea University) research ethics committee (SU-ethics-310118/478).

## Decision letter and Author response

Decision letter https://doi.org/10.7554/eLife.70356.sa1
Author response https://doi.org/10.7554/eLife.70356.sa2

---

# Additional files

## Supplementary files

• Supplementary file 1. Additional data analysis. Linear regression of *Hematodinium* intensity in the liquid tissue (parasites per mL haemolymph) log transformed [Y = log(y + 1)] (liquid tissues) against average histology severity rating from gills and hepatopancreas (solid tissues) for infection severity (n = 108).

• Supplementary file 2. Statistical outputs from full binomial logistic regression models using *Hematodinium*-positive crabs determined via PCR alone n = 162 (table 1, models S1–S6) and models run using crabs with clinical signs of *Hematodinium* sp. (n = 111) determined via PCR as well as liquid/sold tissue inspections, and intensity of *Hematodinium* sp. infection (table 2, models S7–S14).

• Supplementary file 3. Additional data analysis. Non-metric multidimensional (nMDS) ordination collateral infection community structure(s) in crabs that were *Hematodinium* sp. positive (orange) for clinical signs of disease (diagnosed by PCR as well as liquid/tissue inspection), and *Hematodinium* sp. free (black – control), and collateral infection community structure(s) in crabs from the Dock (red) and Pier (blue) locations.

• Supplementary file 4. Complete list of BLAST search results and genetic identifiers for all 'collateral infections' amplified via PCR and sequenced using Sanger's method.

• Transparent reporting form

• Source data 1. Data underpinning Figures 2–5.

• Source code 1. ('R') for BLR models.

• Source code 2. ('R') for revised BLR models (excluding subclinical).

## Data availability

Sequencing data have been deposited in GenBank under accession codes numbers MN846355 - MN846359 (from *C. maenas* haemolymph DNA) and MT334463 - MT334513 (seawater eDNA) for haplosporidia (Davies et al., 2020a); MN985606 - MN985608 (seawater eDNA) and MN985609 (*C. maenas* haemolymph DNA) for microsporidia, MN985610 - MN985644 (seawater eDNA) for paramyxids, MT000071 - MT000098 (seawater eDNA) for mikrocytids and MT000100 - MT000103 (*C. maenas* haemolymph DNA) and MT000104 - MT000107 (seawater eDNA) for fungal species. Sequences for *Vibrio* spp. (<150 bp in length) were deposited in the NCBI short read archive (SRA) under accession numbers SAMN14133753 to SAMN14133757 (*C. maenas* haemolymph DNA) and SAMN14133758- SAMN14133765 (seawater eDNA; see supplementary files for complete information). All source files (code, data, supporting information) are available immediately, and have been submitted alongside the manuscript.

The following dataset was generated:

| Author(s) | Year | Dataset title | Dataset URL | Database and Identifier |
| --- | --- | --- | --- | --- |
| Davies CE, Thomas JE, Malkin SH, Batista FM, Rowley AF, Coates CJ | 2020 | *Vibrio* spp. in *Carcinus maenas* | https://www.ncbi.nlm.nih.gov/bioproject/PRJNA607439 | NCBI BioProject, PRJNA607439 |

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

# Appendix 1

**Appendix 1—table 1.** Genetic diversity indices of *C. maenas* COI sequences (481 bp).

| Location | Code | N | H | P | Hd (SD) | π (SD) |
|---|---|---|---|---|---|---|
| Seltjarnarnes, Iceland | ICE | 18 | 1 | 0 | 0 | 0 |
| Torshavn, Faroe Islands | TOR | 19 | 2 | 1 | 0.515 (0.052) | 0.0032 (0.0022) |
| Mongstad, Norway | MON | 22 | 12 | 5 | 0.853 (0.065) | 0.0044 (0.0028) |
| Oslo, Norway | OSL | 9 | 5 | 1 | 0.806 (0.120) | 0.0067 (0.0043) |
| Goteborg, Sweden | GOT | 15 | 10 | 3 | 0.933 (0.045) | 0.0067 (0.0041) |
| Den Helder, the Netherlands | NET | 45 | 17 | 7 | 0.784 (0.059) | 0.0043 (0.0027) |
| Dock, Swansea, UK | DOCK | 45 | 18 | 8 | 0.855 (0.038) | 0.0042 (0.0027) |
| Mumbles Pier, UK | PIER | 48 | 19 | 10 | 0.823 (0.041) | 0.0039 (0.0025) |
| Fowey, England | FOW | 14 | 8 | 3 | 0.890 (0.060) | 0.0047 (0.0031) |
| Bilbao, Spain | BIL | 15 | 6 | 3 | 0.648 (0.134) | 0.0031 (0.0022) |
| Aveiro, Portugal | AVE | 23 | 9 | 3 | 0.795 (0.065) | 0.0037 (0.0025) |
| Cádiz, Spain | CAD | 47 | 21 | 12 | 0.864 (0.042) | 0.0039 (0.0025) |

N = number of individuals analysed; H = number of haplotypes; P = private haplotypes; Hd (SD) = haplotype diversity (standard deviation); π (SD), nucleotide diversity (standard deviation).

**Appendix 1—table 2.** Pairwise genetic differentiation (Fst) of *C. maenas* samples estimated using COI sequence data.

| | 1 ICE | 2 TOR | 3 MON | 4 OSL | 5 GOT | 6 NET | 7 DOCK | 8 PIER | 9 FOW | 10 BIL | 11 AVE | 12 CAD |
|---|---|---|---|---|---|---|---|---|---|---|---|---|
| 2 | 0.381 | 0.000 | | | | | | | | | | |
| 3 | 0.839 | 0.688 | 0.000 | | | | | | | | | |
| 4 | 0.859 | 0.670 | 0.022 | 0.000 | | | | | | | | |
| 5 | 0.789 | 0.604 | 0.013 | –0.024 | 0.000 | | | | | | | |
| 6. | 0.802 | 0.678 | 0.019 | 0.064 | 0.038 | 0.000 | | | | | | |
| 7 | 0.797 | 0.666 | 0.028 | **0.083** | 0.016 | 0.001 | 0.000 | | | | | |
| 8 | 0.805 | 0.675 | **0.070** | **0.123** | 0.045 | 0.010 | –0.006 | 0.000 | | | | |
| 9 | 0.857 | 0.675 | –0.016 | 0.015 | –0.018 | –0.019 | –0.012 | 0.003 | 0.000 | | | |
| 10 | 0.902 | 0.730 | 0.054 | 0.118 | 0.067 | –0.018 | 0.006 | 0.007 | 0.012 | 0.000 | | |
| 11 | 0.859 | 0.710 | –0.006 | 0.068 | 0.040 | –0.013 | 0.001 | 0.027 | –0.011 | –0.012 | 0.000 | |
| 12 | 0.821 | 0.710 | 0.032 | **0.108** | **0.103** | 0.005 | **0.042** | **0.057** | 0.016 | –0.009 | –0.006 | 0.000 |

Values in bold are statistically significant (p<0.05).

**Appendix 1—table 3.** Genetic diversity indices of *Hematodinium* sp.
ITS-1 sequences from infected *C. maenas* collected in the Swansea Dock and in the Mumbles Pier during winter, spring, summer, and autumn.

| Location | Season | n | h | Hd (SD) | π (SD) |
|---|---|---|---|---|---|
| Pier | Winter | 8 | 3 | 0.464 (0.200) | 0.0305 (0.0182) |
| | Spring | 20 | 15 | 0.958 (0.033) | 0.0180 (0.0105) |
| | Summer | 13 | 11 | 0.961 (0.050) | 0.0201 (0.0123) |
| | Autumn | 8 | 8 | 1.000 (0.0625) | 0.0191 (0.0120) |
| | Total | 49 | 31 | 0.917 (0.0343) | 0.0199 (0.0109) |
| Dock | Winter | 13 | 9 | 0.872 (0.091) | 0.0322 (0.0181) |
| | Spring | 14 | 11 | 0.934 (0.061) | 0.0261 (0.0149) |
| | Summer | 15 | 15 | 1.000 (0.024) | 0.0289 (0.0162) |
| | Autumn | 11 | 10 | 0.982 (0.046) | 0.0274 (0.0159) |
| | Total | 53 | 41 | 0.951 (0.025) | 0.0274 (0.0146) |
| Both sites | Overall | 102 | 70 | 0.935 (0.021) | 0.0228 (0.0122) |

n = number of individuals analysed; h = number of haplotypes; P = private haplotypes; Hd (SD) = haplotype diversity (standard deviation); π (SD), nucleotide diversity (standard deviation).

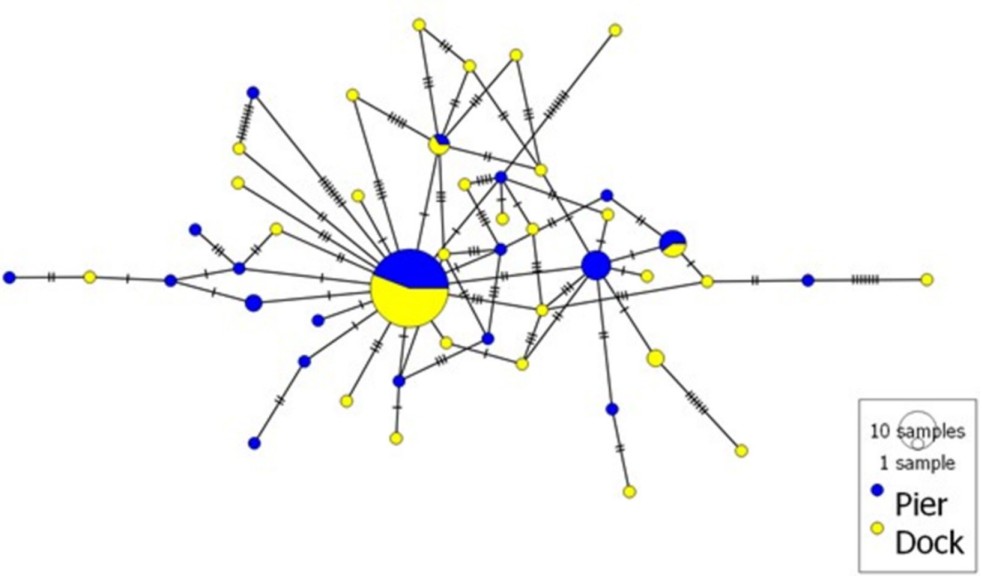

**Appendix 1—figure 1.** Haplotype network of partial ITS region (252 bp) from *Hematodinium* species infecting shore crabs. In total, 102 *Hematodinium* sequences from two sites (n = 49/Pier; n = 53/Dock) were analysed. The size of each circle depicted is proportional to the frequency of a haplotype within the dataset. The network was visualised using POPART (***Leigh and Bryant, 2015***).

# Appendix 2

**Appendix 2—table 1.** Forward (Fwd) and reverse (Rev) primer sequences used for the amplification of pathogens from *C. maenas* and host DNA, by PCR.

Each PCR run included initial denaturation and final extension steps, according to the first and final temperatures, respectively, noted in the thermocycler settings.

| Target pathogen (Reference) | Primers | | Cycling conditions | | |
|---|---|---|---|---|---|
| | Dir. \| Name \| Sequence (5'–3') | Final conc. (µM) | Temp (°C) | Time | Cycles |
| *Hematodinium* sp. (*Gruebl et al., 2002*) | For, HematF1487: | 0.5 | 94 | 10 min | 30 |
| | CCTGGCTCGATAGAGTTG | | 94 | 15 s | |
| | Rev, HematR1654: | | 54 | 15 s | |
| | GGCTGCCGTCCGAATTATTCAC | | 72 | 30 s | |
| | [Amplicon size, 187 bp] | | 72 | 10 min | |
| Fungi (*Gardes and Bruns, 1993*; *White et al., 1990*) | For, ITS1F: | 0.4 | 95 | 2 min | 30 |
| | CTTGGTCATTTAGAGGAAGTAA | | 95 | 30 s | |
| | Rev, ITS2: | | 55 | 30 s | |
| | GCTGCGTTCTTCATCGATGC | | 72 | 1 min | |
| | [Amplicon size, 320 bp] | | 72 | 10 min | |
| *Haplosporidia* spp. (round 1) (*Hartikainen et al., 2014a*) | For, C5fHap: | 1 | 95 | 5 min | 30 |
| | GTAGTCCCARCYATAACBATGTC | | 95 | 30 s | |
| | Rev, Sb1n: | | 65 | 45 s | |
| | GATCCHTCYGCAGGTTCACCTACG | | 72 | 1 min | |
| | [Amplicon size, NA bp] | | 72 | 10 min | |
| *Haplosporidia* spp. (round 2) (*Hartikainen et al., 2014a*) | For, V5fHapl: | 1 | 95 | 5 min | 30 |
| | GGACTCRGGGGGAAGTATGCT | | 95 | 30 s | |
| | Rev, Sb2nHap: | | 65 | 45 s | |
| | CCTTGTTACGACTTBTYCTTCCTC | | 72 | 1 min | |
| | [Amplicon size, 650 bp] | | 72 | 10 min | |
| Mikrocytids (round 1) (*Hartikainen et al., 2014b*) | For, mik451F: | 0.5 | 95 | 5 min | 30 |
| | GCCGAGAYGGTTAAWGAGCCTCCT | | 95 | 30 s | |
| | Rev, mik1511R: | | 64 | 45 s | |
| | CCTATTCAGCGCGCTCTGTTGAGA | | 72 | 1 min | |
| | [Amplicon size, 967 bp] | | 72 | 10 min | |
| Mikrocytids (round 2) (*Hartikainen et al., 2014b*) | For, mik868F: | 0.5 | 95 | 5 min | 30 |
| | GGACTACCAGWGGCGAAAGCGCCT | | 95 | 30 s | |
| | Rev, mik1340R: | | 62 | 45 s | |
| | TGCATCACGGACCTACCTTWGACC | | 72 | 1 min | |
| | [Amplicon size, 481 bp] | | 72 | 10 min | |

*Appendix 2—table 1 Continued on next page*

*Appendix 2—table 1 Continued*

| Target pathogen (Reference) | Primers | | Cycling conditions | | | |
|---|---|---|---|---|---|---|
| *Vibrio* spp. (*Thompson et al., 2004*; *Vezzulli et al., 2012*) | For, 567 F: | 0.5 | 94 | 10 min | 30 | |
| | GGCGTAAAGCGCATGCAGGT | | 94 | 30 s | | |
| | Rev, 680 R: | | 58 | 30 s | | |
| | GAAATTCTACCCCCCTCTACAG | | 72 | 1 min | | |
| | [Amplicon size, 113 bp] | | 72 | 10 min | | |
| Microsporidia (*Fedorko et al., 1995*; *Stentiford et al., 2018*) | For, CTMicrosp-G: | 0.5 | 94 | 5 min | 35 | |
| | CACCAGGTTGATTCTGCCTGAC | | 94 | 30 s | | |
| | Rev, Microsp1342r: | | 63 | 30 s | | |
| | ACGGGCGGTGTGTACAAAGAACAG | | 72 | 90 s | | |
| | [Amplicon size, 1100–1300 bp] | | 72 | 10 min | | |
| Paramyxid (round 1) (*Ward et al., 2016* ) | For, Para1+ fN: | 1 | 95 | 3 min | 42 | |
| | GCGAGGGGTAAAATCTGAT | | 95 | 30 s | | |
| | Rev, ParaGENrDBn: | | 67 | 1 min | | |
| | GTGTACAAAGGACAGGGACT | | 72 | 1 min | | |
| | [Amplicon size, NA bp] | | 72 | 5 min | | |
| Paramyxid (round 2) (*Ward et al., 2016*) | For, Para3+ fN: | 1 | 95 | 3 min | 42 | |
| | GGCTTCTGGGAGATTACGG | | 95 | 30 s | | |
| | Rev, Para2+ rN: | | 62 | 1 min | | |
| | TCGATCCCRACTGRGCC | | 75 | 1 min | | |
| | [Amplicon size, 450 bp] | | 75 | 5 min | | |
| Cytochrome c oxidase I (COI) gene (Roman and Palumbi, 2004) | For, Cm_F: | 0.5 | 94 | 2 min | 30 | |
| | GCTTGAGCTGGCATAGTAGG | | 94 | 1 min | | |
| | Rev, Cm_R: | | 50 | 1 min | | |
| | GAATGAGGTGTTTAGATTTCG | | 72 | 1 min | | |
| | [Amplicon size, 588 bp] | | 72 | 10 min | | |

