## [Editor Report]

Davies et al. present a large-scale field survey of infection status in crabs. *Hematodinium* sp., a dinoflagellate parasite that impacts crustacean fisheries worldwide. *Hematodinium* sp., previously thought to render crabs more susceptible to other infectious agents via immunosuppression, was not found to be associated with collateral infections with other disease agents. This study, instead, presents a new framework for *Hematodinium*-crab interactions; latency of the infection and absence of host immune response may drive the endemic status of *Hematodinium* sp. infections in crustaceans.

---

## [Decision Letter]

**Decision letter after peer review:**

Thank you for submitting your article "Environment, rather than Hematodinium parasitization, determines collateral disease contraction in a crustacean host" for consideration by *eLife*. Your article has been reviewed by 3 peer reviewers, including Irene Salinas as the Reviewing Editor and Reviewer #3, and the evaluation has been overseen by Carla Rothlin as the Senior Editor.

Essential revisions:

Reviewers agreed that additional experiments are essential to support the main conclusions of this work.

1) Present quantitative Hematodinium pathogen loads using molecular techniques (i.e qPCR) that complement the histological diagnostic. qPCR may be carried in out in a subset of samples if not all of them are available for molecular diagnosis. The new data should be then analyzed in the context of presence of co-infections.

2) The immune evasion hypothesis is not supported by the data presented in the manuscript. Thus, validation or further support for the immune evasion hypothesis is necessary to strengthen the data shown in Figure 6. Additional experiments needs to be quantitative. For instance, authors may perform time series experiments of phagocytosis assays in vitro with different life stages of the parasite. These assays should be accompanied by the necessary quantification of these experiments. Additionally, a time course study infecting naive crabs with the parasite and following the immune response over time (humoral and cellular) needs to be included in order to rule out that the host mounts immune responses against Hematodinium sp.. These experiments will shed light to the currently unknown immune response to this parasite. An elegant way to further substantiate these findings would be to perform proteomics of hemolymph samples from uninfected and infected crabs over time.

*Reviewer #1 (Recommendations for the authors):*

Overall, I think the authors made a solid effort to test their hypothesis; however, I believe data is needed to prove or disprove it. I think the discussion and the second conclusion could be accordingly modified.

*Reviewer #2 (Recommendations for the authors):*

1. Lines 320-323: Although there were no significant differences between the number of disease-agents between Hematodinium-positive and Hematodinium-free crabs, irrespective of the two locations examined, regardless of location, the p value is clone to 0.9. It appears is a trend in the difference in pathogen prevalence between Hematodinium-positive and Hematodinium-free crabs. A metagenomic analysis (targeting microbial pathogens of crab) might reveal a significant difference between the two population that could be reveled when the two populations are screened for a targeted set of pathogens by PCR. This could eb a serious limitation of this sort of study, as PCR-based diagnosis is focused on a set of pathogens than providing a global view of pathogen prevalence in a population. The fact that the number of bacterial CFUs was significantly higher in the haemolymph of Hematodinium-affected crabs compared to Hematodinium-free crabs in at least one of the two locations elude to the fact that in combination with yet unknown factor (biotic or abiotic), Hematodinium infection perhaps modulate host defense.

2. Lines 462-465: Is it possible that Hematodinium is not passively evading host response, instead produce pathogen encoded effectors that shut down pathogen recognition. So it is more of an active process than a passive evasion by the parasite? In that event, the conclusion made in this manuscript will not be accurate. A large-scale metagenomic study looking into the transcriptome profiles of Hematodinium-infected and healthy shore crab can unequivocally address if there is any global gene expression changes induced by the parasite that manifest in an apparent immune invasion (but in reality an active host immune suppression approach used by the pathogen).

3. Lines 573-578: These observations further elude to the fact that the apparent immune evasion is actually a very active process employed by the pathogen not a passive process at all, as one might speculate from this manuscript.

4. Looked into equal number of Hematodinium-positive crabs and size/sex-matched these to Hematodinium-free crabs (N=162) out of 1,191 analysed. What were the basis of selecting these 162 samples? Was there only 162 animals out of 1191 infected with Hematodinium, and if so what was the basis of selecting presumably 162 healthy animals. This is important because any bias in selection will modulate the downstream analyses and conclusions made.

5. The authors also looked into environmental DNA (eDNA) from the surrounding waters of infected crabs to assess the spatial and temporal ecology of different pathogens. Did the authors collect samples at different time throughout the grow-out period and when crabs are not cultured? T is not apparent from the information presented here.

6. In the abstract, lines 29-31, the authors mentioned about multi-resource approach in screening samples (N=162). But in the method section, they describe determining population diversity looking into 100 animals in total (50 animals in each of the wo sites examined).

*Reviewer #3 (Recommendations for the authors):*

This work has a lot of merit for several reasons, including the importance of the topic, the lack of studies in this area and the large data set from a wide geographical range.

My main issues are the lack of adequate tools and approaches to ascertain the claims and conclusions of the manuscript.

My main criticisms are:

– Lack of quantitative data to evaluate Hematodinium parasite loads. Currently, the authors only provide yes or no using histological diagnosis. For many reasons, having a molecular and quantitative (or semi-quantitative) diagnostic tool would greatly improve the manuscript and help reach much more solid conclusions.

– Immunological assays: the data provided to support the immune evasion hypothesis is very weak and again not quantitative. The fact that the parasites are not seen to be engulfed in ex vivo assays when mixed with healthy hemocytes or that encapsulated Hematodinium cannot be observed in histological sections is not quantitative and requires further investigation and validation. Specifically, regarding phagocytosis:

– It is known that phagocytosis is dependent on the size of the target particle. Thus, the fact that authors did not observe engulfment of parasites in their assays by light microscope does not rule out that phagocytosis of other life stages of the parasite does occur. Further, in a real infection scenario, host-derived factors induced by infection such as complement and cytokines may be critical for the phagocytosis of the parasite and these conditions were not evaluated in this study.

– Immune evasion of cellular immune responses does not mean that, at the molecular level, the parasite triggers host immune responses. Again, this aspect is not measured in this study and therefore the immune evasion theory seems premature in my opinion. A way to strengthen this aspect could be proteomic analysis of hemolymph from control and infected individuals.

– The evidence of lack of encapsulation is based on histological examination which may miss different life stages of the parasite and does not include any immunohistochemistry using anti-Hematodinium specific antibodies. Ideally, the IHC would detect all life stages of the parasite to draw conclusive evidence. I am not sure if these tools are available or not. Alternatively, in situ hybridization would be a very helpful too. The authors just claim that these crabs are still able to encapsulate other agents (but not Hematodinium) and show one image of debris surrounded by hemocytes. In order to support these claims, an experimental infection of healthy crabs with the parasite needs to be performed and the immune response quantified over time using the adequate controls and tools.

Overall, refinement of the diagnostic tools and immunological assays is necessary. Otherwise, the data set used for this study is invaluable and can help us learn more about marine epidemics.

[Editors' note: further revisions were suggested prior to acceptance, as described below.]

Thank you for resubmitting your work entitled "Environment, rather than Hematodinium parasitization, determines collateral disease contraction in a crustacean host" for further consideration by *eLife*. Your revised article has been reviewed by 2 peer reviewers and the evaluation has been overseen by Carla Rothlin as the Senior Editor, and a Reviewing Editor.

The manuscript has been improved but there are some remaining issues that need to be addressed. Please find below the list of points to be addressed in your revision:

– Title: The reviewers and the Reviewing editor consider that the title of the manuscript is premature considering there is not enough environmental factors/ host immunity analyzed to justify that environment is the only driving factor for collateral infection. It is certainly tempting to speculate environment as the driving factor considering lack of pathogen driven factor influencing susceptibility (albeit based on the parameters analyzed). The authors might consider a title more on the empirical evidence they found, e.g. Hematodinium infection does not drive collateral infection.

– The observation that a crab that has suffered limb loss is 2.4-fold less likely to present a co-infection is intriguing. Would that mean that trauma due to limb loss may stimulate the immune system to repair/re-grow and at the same time keep at bay co-infections? Please include in the discussion.

---

## [Author Response]

Essential revisions:Reviewers agreed that additional experiments are essential to support the main conclusions of this work.1) Present quantitative Hematodinium pathogen loads using molecular techniques (i.e qPCR) that complement the histological diagnostic. qPCR may be carried in out in a subset of samples if not all of them are available for molecular diagnosis. The new data should be then analyzed in the context of presence of co-infections.

We have now incorporated quantitative *Hematodinium* data. In our original submission, we presented prevalence data that discriminated between *Hematodinium*-positive crabs (n = 162) and *Hematodinium*-negative crabs (n = 162) based on targeted PCR alone. Rather than use qPCR – due to the absence of sufficient molecular information to ensure single-copy gene targets per genome for *Hematodinium* spp. and the diversity of ecotypes presented in our study – we have incorporated count data (actual number of parasites per mL haemolymph) from the original liquid tissue screens using haemocytometry, and grade data (0-4) from solid (gill and hepatopancreas) tissues assessed using histology.

We have analysed these data in the context of co-infection presence. Indeed, when we restricted the *Hematodinium*-positive crabs to n = 111 based on individuals that were positive for the parasite in all three diagnostic methods across key tissues (haemolymph via haemocytometry and PCR, and gill/hepatopancreas via histology), and re-ran *all* our analyses/models, the outcomes did not yield contradictory conclusions. *Hematodinium-*positive crabs are no more likely to contain co-infections when compared to *Hematodinium*-negative crabs, AND, location remains the determining factor for collateral infections (pathogen) among crabs.

2) The immune evasion hypothesis is not supported by the data presented in the manuscript. Thus, validation or further support for the immune evasion hypothesis is necessary to strengthen the data shown in Figure 6. Additional experiments needs to be quantitative. For instance, authors may perform time series experiments of phagocytosis assays in vitro with different life stages of the parasite. These assays should be accompanied by the necessary quantification of these experiments. Additionally, a time course study infecting naive crabs with the parasite and following the immune response over time (humoral and cellular) needs to be included in order to rule out that the host mounts immune responses against Hematodinium sp.. These experiments will shed light to the currently unknown immune response to this parasite. An elegant way to further substantiate these findings would be to perform proteomics of hemolymph samples from uninfected and infected crabs over time.

We accept the need to be more circumspect regarding the presentation/interpretation of our data, but the assay was quantitative and assessed haemocyte (immune cell) reactivity beyond phagocytosis, i.e., encapsulation. Several slides of freshly withdrawn haemolymph and processed gill/hepatopancreas were inspected via microscopy, which amounts to a minimum of 486 slides for the 162 Hematodinium-positive samples. Per crab, 1 slide of freshly withdrawn haemolymph, and 4-5 slides of multi-tissue histology were viewed, graded (according to Smith et al., 2015) and photographed. These showed *conclusively* that there is/was no evidence of cellular host reactivity to the *Hematodinium* morphotypes.

In July 2021, we screened a further 58 crabs for *Hematodinium*. For the positive animals (n = 9), we maintained the haemolymph ex vivo for up to 75 minutes. No discernible haemocyte response to the parasites was observed. In a separate experiment, we separated haemocytes from six healthy (*Hematodinium*-free) crabs and incubated them in vitro with *Hematodinium* isolated from an infected donor crab (following Smith and Rowley, 2015, and Coates et al., 2012) in wells of a microplate for 2 hours. Again, *no haemocyte reactivity* to these parasites was observed.

In a previous study concerning another species of crab (*Cancer pagurus*), Smith and Rowley (2015) infected crabs with 1 x10^5^ trophonts of *Hematodinium* sp. and monitored crab health, including haemocyte (immune cells) over 1 year. No phagocytosis or encapsulation of *Hematodinium* was observed, and mortality events after 90 days in both *Hematodinium*-positive and *Hematodinium*-negative crabs could be attributed to a range of other parasites, including bacteria and mikrocytids – agents that we have targeted systematically in our current study.

We are looking to investigate proteomic signatures from *Hematodinium*-positive versus *Hematodinium*-free shore crabs, but we cannot perform that analysis on the animals used in this current study as the haemolymph was spent for bacterial CFU screening, *Hematodinium* and haemocyte enumeration, and DNA extraction for PCR diagnostics.

Reviewer #1 (Recommendations for the authors):Overall, I think the authors made a solid effort to test their hypothesis; however, I believe data is needed to prove or disprove it. I think the discussion and the second conclusion could be accordingly modified.

We thank Reviewer 1 for their time and insight. We have already answered the comment on data analyses. We have revised the discussion and conclusion to reflect the comments from this referee, and the more cautious interpretation of our study.

Reviewer #2 (Recommendations for the authors):1. Lines 320-323: Although there were no significant differences between the number of disease-agents between Hematodinium-positive and Hematodinium-free crabs, irrespective of the two locations examined, regardless of location, the p value is clone to 0.9. It appears is a trend in the difference in pathogen prevalence between Hematodinium-positive and Hematodinium-free crabs. A metagenomic analysis (targeting microbial pathogens of crab) might reveal a significant difference between the two population that could be reveled when the two populations are screened for a targeted set of pathogens by PCR. This could eb a serious limitation of this sort of study, as PCR-based diagnosis is focused on a set of pathogens than providing a global view of pathogen prevalence in a population. The fact that the number of bacterial CFUs was significantly higher in the haemolymph of Hematodinium-affected crabs compared to Hematodinium-free crabs in at least one of the two locations elude to the fact that in combination with yet unknown factor (biotic or abiotic), Hematodinium infection perhaps modulate host defense.

The P-values reported here are not from correlative analysis. There is no trend to suggest pathogen load differences/diversity between *Hematodinium*-positive versus *Hematodinium*-negative crabs. The study was not limited to PCR alone but covered disease histologically and via microscopic examination of haemolymph (approaches that have determined the presence/absence of haplosporidians, fungi, trematodes and *S. carcini* in shore crabs and were cited throughout our text).

The reviewer is correct in that performing a metagenomic approach would likely reveal a more complex pathobiont and symbiont profile of the crabs, but would not change the overall conclusion of our study that *Hematodinium* parasitisation does not increase a crabs’ likelihood of contracting additional diseases. This was the main aim of our study, and addresses a misconception of *Hematodinium* infection dynamics in the literature.

Crabs from the docks do contain higher CFU counts but this is attributed to the presence of *S. carcini* infection. Indeed, we cited our published article in which *S. carcini*-positive crabs contain significantly higher CFUs when compared to *S. carcini*-free crabs. Although Figure 3g displays this data clearly, when running our Binomial Logistic Regression models (Table1 and Supplementary Data), CFUs did not come out as a significant predictor variable.

2. Lines 462-465: Is it possible that Hematodinium is not passively evading host response, instead produce pathogen encoded effectors that shut down pathogen recognition. So it is more of an active process than a passive evasion by the parasite? In that event, the conclusion made in this manuscript will not be accurate. A large-scale metagenomic study looking into the transcriptome profiles of Hematodinium-infected and healthy shore crab can unequivocally address if there is any global gene expression changes induced by the parasite that manifest in an apparent immune invasion (but in reality an active host immune suppression approach used by the pathogen).

This is an interesting point. We do not suggest in our manuscript that *Hematodinium* is passively evading the immune response – rather we consider *Hematodinium* is perhaps disguising itself in a manner (but we do not speculate further – experimental work is on-going). If it was actively producing effectors that shut down recognition – this is immune suppression, and we would expect to see higher susceptibility to infection (we don’t), and a reduced host response to other infectious agents (again, we don’t). We are in the process of looking at proteomic responses – but another group has looked at *Hematodinium* in an alternative host and did not identify *Hematodinium*-derived factors, yet did identify differentially protein expression in the hepatopancreas of parasitised versus non-parasitised portunid crabs (Li et al., 2019 – cited in our paper).

3. Lines 573-578: These observations further elude to the fact that the apparent immune evasion is actually a very active process employed by the pathogen not a passive process at all, as one might speculate from this manuscript.

Immune evasion is not a passive process, and we do not suggest this in our paper (the word passive is never mentioned). Many parasites, including pathogenic fungi, can shield their surface ligands from pathogen recognition receptors and opsonins – thereby avoiding an immune response without the need to suppress it. Our revised manuscript is more cautious in this regard.

4. Looked into equal number of Hematodinium-positive crabs and size/sex-matched these to Hematodinium-free crabs (N=162) out of 1,191 analysed. What were the basis of selecting these 162 samples? Was there only 162 animals out of 1191 infected with Hematodinium, and if so what was the basis of selecting presumably 162 healthy animals. This is important because any bias in selection will modulate the downstream analyses and conclusions made.

We have expanded upon this in our text. 1,191 animals were screened across 12 months. 162 tested positive for *Hematodinium* via PCR, and were also assessed microscopically/histologically. Of the remaining ~1000 *Hematodinium*-negative crabs, 162 were matched based on size, location and sex, and were also further processed histologically. This approach ensured parity.

5. The authors also looked into environmental DNA (eDNA) from the surrounding waters of infected crabs to assess the spatial and temporal ecology of different pathogens. Did the authors collect samples at different time throughout the grow-out period and when crabs are not cultured? T is not apparent from the information presented here.

These crabs are not cultured. The sample sites in Swansea Bay are from natural populations with no anthropogenic intervention for cultivation, or known commercial activity. eDNA samples were collected on the same sampling days as crabs.

6. In the abstract, lines 29-31, the authors mentioned about multi-resource approach in screening samples (N=162). But in the method section, they describe determining population diversity looking into 100 animals in total (50 animals in each of the wo sites examined).

We apologise for any unintentional confusion. 1191 crabs were screened across two sites for 1 year. 162 animals that tested positive for *Hematodinium*, plus 162 matched *Hematodinium* negative crabs were processed for ‘collateral infections’ to test our main hypothesis. A sub sample (n = 100) of the 1191 crabs were processed for genetic population diversity between the two locations.

Reviewer #3 (Recommendations for the authors):As mentioned earlier, this work has a lot of merit for several reasons, including the importance of the topic, the lack of studies in this area and the large data set from a wide geographical range.My main issues are the lack of adequate tools and approaches to ascertain the claims and conclusions of the manuscript.My main criticisms are:– Lack of quantitative data to evaluate Hematodinium parasite loads. Currently, the authors only provide yes or no using histological diagnosis. For many reasons, having a molecular and quantitative (or semi-quantitative) diagnostic tool would greatly improve the manuscript and help reach much more solid conclusions.

This is not entirely accurate – the presence/absence of *Hematodinium* was first assessed via haemolymph microscopic inspection, followed by PCR-based diagnosis to account for any sub-clinical presence. If positive via PCR (n = 162), crabs were further assessed via multi-tissue histology. In the revised text, we have added the actual number of *Hematodinium* recorded (parasite load per mL) of the haemolymph, and the grade of infection (0 – 4) for both gill and hepatopancreas (methods have been published previously and cited in our main text).

– Immunological assays: the data provided to support the immune evasion hypothesis is very weak and again not quantitative. The fact that the parasites are not seen to be engulfed in ex vivo assays when mixed with healthy hemocytes or that encapsulated Hematodinium cannot be observed in histological sections is not quantitative and requires further investigation and validation. Specifically, regarding phagocytosis:

Admittedly, we should have been more circumspect in our conclusions, however, our approach was at least semi-quantitative. Several thousand slides were inspected, and no gross cellular responses were recorded. To reinforce this point, we performed two additional ‘proof of concept studies’ (described above).

– It is known that phagocytosis is dependent on the size of the target particle. Thus, the fact that authors did not observe engulfment of parasites in their assays by light microscope does not rule out that phagocytosis of other life stages of the parasite does occur.

This is, of course, an important point. The trophont morphotypes visibly infecting the crabs do not induce immune reactivity at least at the cellular level. Perhaps a smaller life stage is phagocytosed – we did not observe internalised *Hematodinium* in any of our diagnostics, and to the best of our knowledge, there is no mention of any other life stage phagocytosed by a marine invertebrate in the available literature.

This is experimentally problematic due to a lack of long term established cultures of this parasite, and none exist for those infecting shore crabs (to the best of our knowledge). We have experience of infection trials using an alternative decapod from the same region – *Cancer pagurus*. Crabs infected with 1 x10^5^ trophonts (morphotype universally observed infecting crustaceans) and monitored over a 1 year study, did not show haemocyte-based immune reactivity (melanisation, phagocytosis, encapsulation, or nodule formation; Rowley and Smith, 2015).

Further, in a real infection scenario, host-derived factors induced by infection such as complement and cytokines may be critical for the phagocytosis of the parasite and these conditions were not evaluated in this study.

We did not measure cytokine-like or complement-like factors in our study.

– Immune evasion of cellular immune responses does not mean that, at the molecular level, the parasite triggers host immune responses. Again, this aspect is not measured in this study and therefore the immune evasion theory seems premature in my opinion. A way to strengthen this aspect could be proteomic analysis of hemolymph from control and infected individuals.

We appreciate the reviewer’s insight on this matter, and we should have been more circumspect in our interpretation/writing. We are working on the hypothesis that the parasite avoids detection due to cell surface changes. This is an on-going line of work.

We have begun some proteomic work on the *Hematodinium*-crab system but it will not be relevant to this current paper, as the available crab material has been fully processed already, and was not the overall aim of our study.

– The evidence of lack of encapsulation is based on histological examination which may miss different life stages of the parasite and does not include any immunohistochemistry using anti-Hematodinium specific antibodies. Ideally, the IHC would detect all life stages of the parasite to draw conclusive evidence. I am not sure if these tools are available or not. Alternatively, in situ hybridization would be a very helpful too. The authors just claim that these crabs are still able to encapsulate other agents (but not Hematodinium) and show one image of debris surrounded by hemocytes. In order to support these claims, an experimental infection of healthy crabs with the parasite needs to be performed and the immune response quantified over time using the adequate controls and tools.

We would very much like to be in a position to perform such work, but as the reviewer can appreciate, these tools are not developed for this system (and this extends to the majority of work on marine invertebrate systems).

The panel of host responses in the absence/presence of other agents in addition to *Hematodinium* represents pertinent examples. Even in the presence of tissue damage and/or intact pathogens, the host does not respond to *Hematodinium* when responding to other stimuli that share the same mechanisms of recognition. Again, we have been more circumspect in the revision.

Overall, refinement of the diagnostic tools and immunological assays is necessary. Otherwise, the data set used for this study is invaluable and can help us learn more about marine epidemics.

We are grateful to reviewer 3 for their thoughtful examination of our text.

[Editors' note: further revisions were suggested prior to acceptance, as described below.]

The manuscript has been improved but there are some remaining issues that need to be addressed. Please find below the list of points to be addressed in your revision:– Title: The reviewers and the Reviewing editor consider that the title of the manuscript is premature considering there is not enough environmental factors/ host immunity analyzed to justify that environment is the only driving factor for collateral infection. It is certainly tempting to speculate environment as the driving factor considering lack of pathogen driven factor influencing susceptibility (albeit based on the parameters analyzed). The authors might consider a title more on the empirical evidence they found, e.g. Hematodinium infection does not drive collateral infection.

We have revised our title. *‘Hematodinium* sp. infection does not drive collateral disease contraction in a crustacean host’.

– The observation that a crab that has suffered limb loss is 2.4-fold less likely to present a co-infection is intriguing. Would that mean that trauma due to limb loss may stimulate the immune system to repair/re-grow and at the same time keep at bay co-infections? Please include in the discussion.

This is a very interesting point, and we have considered this previously. Limb loss or damage could temporarily ‘prime’ the immune system of an invertebrate like the shore crab, but at this stage, we are reluctant to speculate further.

There is some evidence that autotomy initiates a stress response in crabs (e.g., *Eriocheir sinensis;* Yang et al. 2018). We added a statement, and the corresponding reference, to the discussion.